# Acetylation of a fungal effector that translocates host PR1 facilitates virulence

Jingtao Li[1], Xiaoying Ma[1], Chenyang Wang[1], Sihui Liu[2], Gang Yu[3], Mingming Gao[1], Hengwei Qian[1], Mengjie Liu[1], Ben F Luisi[4], Dean W Gabriel[5], Wenxing Liang[1]*

[1]Engineering Research Center for Precision Pest Management for Fruits and Vegetables of Qingdao, Shandong Province Key Laboratory of Applied Mycology, College of Plant Health and Medicine, Qingdao Agricultural University, Qingdao, China; [2]College of Science and Information, Qingdao Agricultural University, Qingdao, China; [3]School of Agriculture and Biology, Shanghai Jiao Tong University, Shanghai, China; [4]Department of Biochemistry, University of Cambridge, Cambridge, United Kingdom; [5]Department of Plant Pathology, University of Florida, Gainesville, United States

**Abstract** Pathogens utilize a panoply of effectors to manipulate plant defense. However, despite their importance, relatively little is actually known about regulation of these virulence factors. Here, we show that the effector *Fol*-Secreted Virulence-related Protein1 (FolSvp1), secreted from fungal pathogen *Fusarium oxysporum* f. sp. *lycopersici* (*Fol*), directly binds and translocates the tomato pathogenesis-related protein1, SlPR1, from the apoplast outside the plasma membrane to the host nucleus via its nuclear localization signal. Relocation of SlPR1 abolishes generation of the defense signaling peptide, CAPE1, from its C-terminus, and as a consequence, facilitates pathogen invasion of plants. The action of FolSvp1 requires covalent modification by acetylation for full virulence in host tomato tissues. The modification is catalyzed by the *Fol* FolArd1 lysine acetyltransferase prior to secretion. Addition of an acetyl group to one residue, K167, prevents ubiquitination-dependent degradation of FolSvp1 in both *Fol* and plant cells with different mechanisms, allowing it to function normally in fungal invasion. Either inactivation of FolSvp1 or removal of the acetyl group on K167 leads to impaired pathogenicity of *Fol*. These findings indicate that acetylation can regulate the stability of effectors of fungal plant pathogens with impact on virulence.

*For correspondence:
wliang1@qau.edu.cn

Competing interest: The authors declare that no competing interests exist.

## Editor's evaluation

The authors provided strong evidence that the Fusarium oxysporum effector protein FolSpv1 enhances virulence by targeting tomato SlPR1 and preventing the generation of the SlPR1-derived phytocytokine CAPE1, which otherwise positively regulates disease resistance in tomato plants. Strikingly, they show that FolSpv1 translocates SlPR1 from the apoplast back into the nucleus of tomato cells, suggesting a previously unknown mechanism employed by pathogenic microbes.

## Introduction

In their engagement with pathogens, plants have evolved powerful deterrents to infective organisms, including the well-studied multilayered physical barriers, preformed defenses, and innate immune system (*Zhang et al., 2020*; *Zipfel, 2008*). Pathogen infection often induces the expression of pathogenesis-related (PR) genes (*Kong et al., 2020*), and PR1 is one of the most abundantly accumulated plant proteins upon pathogen attack (*Breen et al., 2017*). PR1 is cleaved at its C-terminus to produce a conserved defense signaling peptide, CAPE1, in response to biotic and abiotic stimuli

to enhance plant immunity through the induction of defense-related genes (*Breen et al., 2017*; *Chen et al., 2014*; *Sung et al., 2021*). PR1 also possesses anti-oomycete activity by the sequestration of sterol from pathogens (*Choudhary and Schneiter, 2012*; *Gamir et al., 2016*, *Niderman et al., 1995*; *Woloshuk et al., 1991*). PR1 is therefore an integral component of host defense, and there is strong selection pressure for pathogens to circumvent it during infection (*Breen et al., 2017*). To date, four effectors from different plant pathogens were shown to interact with PR1 proteins, but the functional impacts on virulence are largely unknown (*Breen et al., 2017*; *Sung et al., 2021*).

Increasing evidence suggests that protein post-translational modifications (PTMs) are important mediators of plant-pathogen interactions (*Carabetta et al., 2019*; *Ren et al., 2017*). PTMs range from small chemical modifications (e.g., acetylation) to the addition of complete proteins (e.g., ubiquitination) (*Ren et al., 2017*). Lysine acetylation is a dynamic and reversible PTM, which is regulated by balanced activities of acetyltransferases and deacetylases, respectively (*Narita et al., 2019*). This PTM is frequently identified on histones and occurs widely in non-histone proteins (*Carabetta et al., 2019*; *Ren et al., 2017*). Acetylation affects protein functions through diverse mechanisms, including by regulating protein stability, subcellular localization, protein-protein interactions, and cross-talk with other PTM events (*Narita et al., 2019*). The bacterial effector protein PopP2 was found to be auto-acetylated, and removal of this modification abolishes virulence of *Ralstonia solanacearum* and RRS1-R-mediated immunity in *Arabidopsis* (*Le Roux et al., 2015*; *Sarris et al., 2015*; *Tasset et al., 2010*). Although the acetylomes of several pathogenic plant fungi, including *Fusarium oxysporum* (*Li et al., 2020a*), *Botrytis cinerea* (*Lv et al., 2016*), and *Fusarium graminearum* (*Zhou et al., 2016*), have been determined by mass spectrometry analysis, the function of these modification in virulence remains poorly characterized.

*F. oxysporum* is a root-infecting fungal pathogen causing wilt disease of a wide variety of plants (*Michielse and Rep, 2009*). *F. oxysporum* f. sp. *lycopersici* (*Fol*), the causal agent of *Fusarium* wilt limited to tomato, invades the roots and subsequently colonizes the xylem vessels, thereby compromising water transport resulting in wilting (*Michielse and Rep, 2009*). During colonization of the host xylem vessels, *Fol* secretes small effectors including 14 different "Secreted-in-Xylem" proteins, and some of them play critical roles in determining host specificity and plant immunity (*Gawehns et al., 2015*; *Houterman et al., 2007*; *Lievens et al., 2009*; *Ma et al., 2013*). In previous studies from our laboratory, we identified 32 acetylated effector candidates secreted at the early infection stage by mass spectrometry (*Li et al., 2020a*). However, the function of these candidate effectors in fungal virulence is unknown.

In this study, we show that one such effector, *Fol*-Secreted Virulence-related Protein1 (FolSvp1), is acetylated on one residue, K167, by the fungal lysine acetyltransferase FolArd1 prior to secretion. This simple covalent modification stabilizes FolSvp1 in *Fol* by repressing ubiquitination-dependent degradation of this protein, allowing for its accumulation and secretion. Secreted FolSvp1 directly interacts with and relocates tomato SlPR1 from the apoplast to host nucleus by way of its nuclear localization signal. This translocation of SlPR1 abolishes generation of the CAPE1 peptide, and as a result, the pathogen invades tomato successfully. K167 acetylation also avoids ubiquitination on this residue in planta, preventing degradation of FolSvp1 by the 26S proteasome before entering plant nucleus. These findings provide a clear example in which acetylation both activates and protects an effector, directly facilitating the successful outcome of a fungal plant pathogen invasion.

## Results

### FolSvp1 is a secretory protein acetylated on K167

Our earlier proteomics studies identified FolSvp1 (FOXG_11456) as a putative secreted protein, acetylated on lysine residue 167 (*Li et al., 2020a*). Prediction with SignalP5.0 revealed the presence of a potential signal peptide (SP) on the N-terminus of FolSvp1 (aa 1–19). We therefore performed a yeast signal trap assay to validate the secretory function of the SP (*Yin et al., 2018*). As shown in *Figure 1A*, the YTK12 strain carrying pSUC2-FolSvp1[1-19] was able to grow on YPRAA medium with raffinose as the sole carbon source and reduce the triphenyltetrazolium chloride to the red-colored insoluble triphenylformazan, indicating that the SP of FolSvp1 is functional.

To confirm secretion of this protein, a recombinant FolSvp1-GFP (green fluorescent protein) fusion with or without the SP was expressed in *Fol* using the RP27 constitutive promoter, with GFP serving as

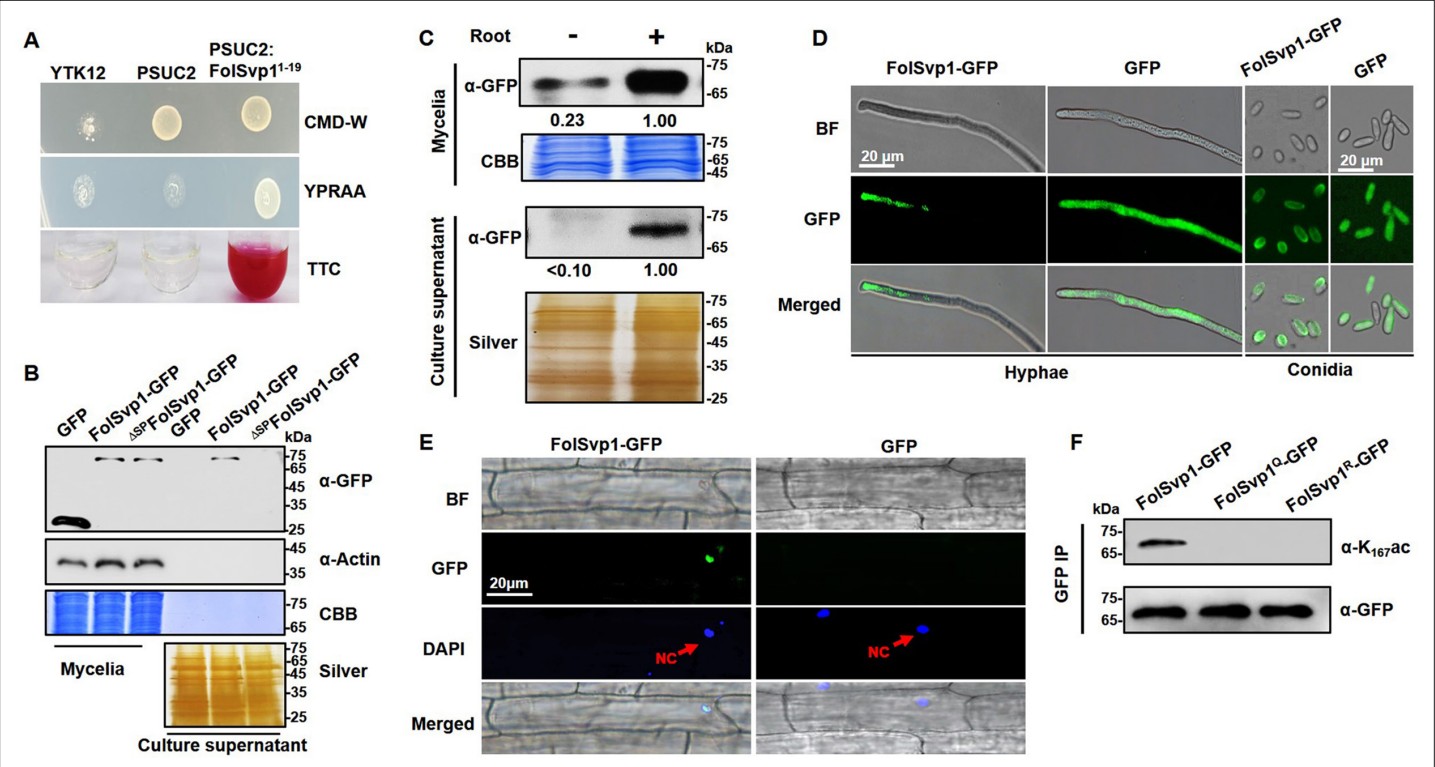

**Figure 1.** Secretion and K167 acetylation of *Fol*-Secreted Virulence-related Protein1 (FolSvp1). (**A**) Functional validation of the signal peptide of FolSvp1. The indicated strains were grown on CMD-W or YPRAA medium for 3 days. Triphenyltetrazolium chloride (TTC) was used to test the enzymatic activity of reducing TTC to red triphenylformazan (TPF). (**B**) Secretion of FolSvp1. Conidia of strains carrying GFP or FolSvp1-GFP with or without signal peptide (SP) (^{ΔSP}FolSvp1-GFP) were inoculated into 5% liquid YEPD in the presence of 2-week-old tomato roots. Forty hours after inoculation, total proteins were extracted from mycelia or culture supernatant, and probed with anti-GFP monoclonal antibody (α-GFP) and anti-Actin antibody (α-Actin), respectively. Coomassie brilliant blue (CBB) or silver staining shows protein loading to each lane. (**C**) Secretion of FolSvp1 with or without tomato roots. The amount of FolSvp1 in mycelia and secreted FolSvp1 in the presence of tomato roots was each set as 1. (**D**) Enrichment of FolSvp1-GFP at hyphal tips. Hyphae and conidia of the indicated strains were imaged. Scale bar, 20 μm. (**E**) Subcellular localization of FolSvp1 in tomato root cells. Three days after inoculation (DAI) by the indicated strains, tomato roots were imaged. Red arrows indicate the nuclei of plants. Scale bar, 20 μm. (**F**) Acetylation of the wild-type (WT) and mutant FolSvp1 proteins. FolSvp1 proteins pulled down from the indicated strains with GFP-Trap beads were probed with α-GFP and anti-K167ac antibody (α-K167ac), respectively. (**D**) and (**F**) were performed in the presence of tomato roots. Each gel shown is a representative experiment carried out three times.

The online version of this article includes the following source data and figure supplement(s) for figure 1:

**Source data 1.** Uncropped images of gels and blots in *Figure 1*.

**Figure supplement 1.** Generation of *Fol*-Secreted Virulence-related Protein1 (*FolSvp1*) mutant strains and phenotypic analysis.

**Figure supplement 1—source data 1.** Uncropped images of gels in *Figure 1—figure supplement 1*.

**Figure supplement 1—source data 2.** Statistical analysis in *Figure 1—figure supplement 1*.

a control (*Figure 1—figure supplement 1A*). Although both FolSvp1-GFP and GFP were expressed in the mycelium, only FolSvp1-GFP with the SP was detected in the culture supernatant induced by tomato roots (*Figure 1B*), indicating that FolSvp1 could be secreted by *Fol* depending on its SP. To further investigate the secretory property of FolSvp1, total proteins were extracted from mycelia and culture supernatant in the presence or absence of tomato roots. As shown in *Figure 1C*, the accumulation and secretion of FolSvp1-GFP was much reduced in the absence of roots. These observations, together with our previous finding that tomato root treatment increases *FolSvp1* transcription (*Li et al., 2020a*), indicate that the presence of tomato roots is critical for the accumulation and secretion of FolSvp1. Compared with the even distribution of GFP protein in the cells, fluorescence microscopic analyses revealed that FolSvp1-GFP was enriched at the apical area of hyphae or conidia (*Figure 1D*), suggesting that FolSvp1 might be secreted through the apical secretion pathway (*Riquelme et al., 2014*; *Steinberg, 2007*). In roots infected with FolSvp1-GFP overexpression strains, GFP signals were

observed in the nucleus of root cells at the inoculation sites (*Figure 1E*). Based on these results, we conclude that secreted FolSvp1 is delivered into plant cells with the nucleus being the accumulation site.

To confirm acetylation of FolSvp1, FolSvp1-GFP was pulled down with anti-GFP agarose beads. Using a specific antibody directed against acetylated K167 of FolSvp1 (anti-K$_{167}$ac), we examined the modification status of FolSvp1 in mycelia. The Western blot analyses detected signals only with the wild-type (WT) FolSvp1 but not with the glutamine (Q) and arginine (R) substitution mutants at position 167 (*Figure 1F*), which mimic the acetylated and unacetylated forms of lysine, respectively, with respect to residue charge (*Carabetta et al., 2019*), indicating that K167 is indeed acetylated prior to secretion.

## K167 acetylation is required for FolSvp1 secretion and full virulence of *Fol*

Although FolSvp1 has homologs in different classes of Pezizomycotina subphylum (*Li et al., 2020a*), phylogenetic analysis showed that it was closely related to members of the *Fusarium* genus (*Figure 2—figure supplement 1A*). Prediction of sequence patterns with the MEME (Multiple Em for Motif Elicitation) tool identified three putative motifs of FolSvp1 and K167 was localized in motif 1 (*Figure 2—figure supplement 1B*). Further alignment of motif 1 showed that the amino acid at 167 site was highly conserved with K or Q residue (*Figure 2—figure supplement 1C and D*) in FolSvp1 homologs, and only two R variations were found in nonpathogenic endophytic fungi *F. oxysporum* Fo47 and *Fusarium austroafricanum* (*Aimé et al., 2013*; *Jacobs-Venter et al., 2018*). These findings suggest that K167 acetylation might be conserved, and associated with pathogenicity of these fungi.

To explore the effect of acetylation on FolSvp1, we expressed mutated versions K167 to Q or R with C-terminal GFP fusions under the control of the RP27 promoter (*Figure 1—figure supplement 1A*). Although no obvious difference was observed in conidia among different transformants, compared with the WT FolSvp1 and the Q mutant, the R mutant displayed weaker fluorescence (*Figure 2A*) and ~75% lower accumulation in mycelia (*Figure 2B*). As a consequence, secretion of the R mutant FolSvp1 was dramatically decreased (*Figure 2B*), and no FolSvp1$^R$ protein was detected in the nucleus of root cells infected with this strain (*Figure 2C and D*). Note that K167 of FolSvp1 secreted into tomato root cells was still acetylated (*Figure 2D*). These results support a critical role of K167 acetylation in FolSvp1 accumulation and secretion, and indicate a function in planta.

To determine the virulence contribution of FolSvp1, we generated *FolSvp1* knockout (Δ*FolSvp1*) mutants and genetic complementation strains (*Figure 1—figure supplement 1B and C*). Although *FolSvp1* deletion did not affect vegetative growth (*Figure 1—figure supplement 1D*) and conidia production (*Figure 1—figure supplement 1E*), virulence of the Δ*FolSvp1* strains was greatly reduced on tomato seedlings (*Figure 2E and F*). Complementation with *FolSvp1* (Δ*FolSvp1-C*) completely restored the virulence phenotype of the Δ*FolSvp1* mutants, indicating that *FolSvp1* is essential for the full virulence of *Fol*. Since K167 acetylation is required for FolSvp1 secretion, it is likely that this modification is physiologically important for *Fol* infection. Indeed, substitution of arginine (R) for K167 (Δ*FolSvp1-C$^R$*) impaired virulence of *Fol*, while mutation of this residue to glutamine (Q) (Δ*FolSvp1-C$^Q$*) had little effect on its infectivity (*Figure 2E and F*). Overall, these data indicate that K167 acetylation of FolSvp1 is likely a key evolutionary step in preparing FolSvp1 to contribute to the virulence of *Fol*.

## K167 acetylation prevents ubiquitination-dependent degradation of FolSvp1 in *Fol*

One possible explanation for the decreased accumulation and secretion of the mutant FolSvp1$^R$ is that K167 deacetylation leads to instability of this protein. To examine this possibility, conidia of FolSvp1$^R$-GFP overexpression strain was cultured for 14 hr and then treated with the proteasome inhibitor MG132 for another 2 hr. With the addition of MG132, FolSvp1$^R$-GFP was elevated ~4-fold and its putatively ubiquitinated form could now be detected (*Figure 3A*). In contrast, MG132 treatment had little effect on FolSvp1-GFP and FolSvp1$^Q$-GFP, indicating that the WT protein was largely acetylated in the presence of tomato root. Western blot analyses using the ubiquitin antibody confirmed ubiquitination of FolSvp1$^R$, but not FolSvp1-GFP and FolSvp1$^Q$-GFP (*Figure 3B*), suggesting that acetylation on K167 prevents ubiquitination and 26S proteasome-dependent degradation of FolSvp1 in *Fol*.

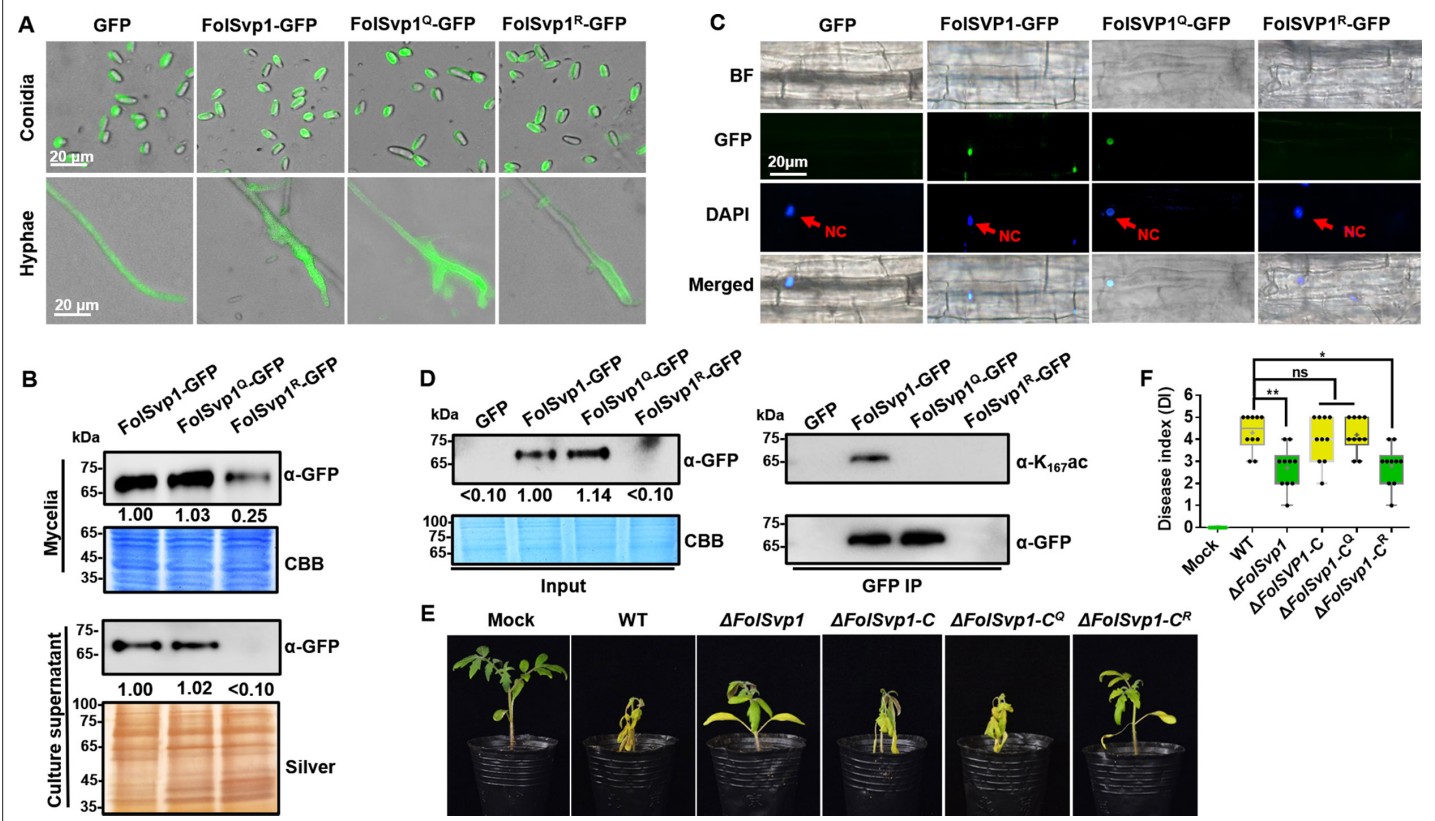

**Figure 2.** Contribution of K167 acetylation to *Fol*-Secreted Virulence-related Protein1 (FolSvp1) secretion and *Fusarium oxysporum* f. sp. *lycopersici* (*Fol*) virulence. (**A**) Location of the wild-type (WT) and mutant FolSvp1 proteins in conidia and hyphae. Scale bar, 20 µm. (**B**) Level of the WT and Q, R mutant FolSvp1 proteins in mycelia and their secreted amount. The amount of WT FolSvp1 in mycelia and secreted was each set as 1. (**C**) Subcellular localization of the WT and Q, R mutant FolSvp1 proteins in tomato root cells. Red arrows indicate the nucleus. Scale bar, 20 µm. For (**A–C**), location, amount, secretion, and subcellular localization of FolSvp1 proteins were determined as in *Figure 1*. (**D**) Amount and acetylation of the WT and Q, R mutant FolSvp1 proteins in tomato root cells. Nuclear proteins were extracted from protoplast of tomato root cells and immunoprecipitated with GFP-Trap beads. FolSvp1-GFP proteins pulled down were then probed with α-GFP and α-K167ac, respectively (right). Input proteins were shown by Western blotting with α-GFP and CBB staining (left). The amount of WT FolSvp1 was set as 1. (**E**) Virulence of the WT and *FolSvp1* mutant strains on tomato. Two-week-old tomato seedlings were inoculated with conidial suspension ($5 \times 10^6$ spores/ml) of the indicated strains. The disease symptoms were observed, and photographs were taken at 14 days after inoculation (DAI). Mock, inoculation with water. (**F**) Disease index scored at 14 DAI. Star represents significant differences according to one-way analysis of variance (ANOVA) (**p<0.01, *p<0.05; ns, no significance; n=10). Whiskers of the boxplots display the upper and lower quartiles, the boxes display the interquartile range, and the plus displays the mean. Each gel shown is a representative experiment carried out three times.

The online version of this article includes the following source data and figure supplement(s) for figure 2:

**Source data 1.** Uncropped gels and blots in *Figure 2*.

**Source data 2.** Statistical analysis in *Figure 2*.

**Figure supplement 1.** Phylogenetic and structural analyses of *Fol*-Secreted Virulence-related Protein1 (FolSvp1) homologs.

**Figure supplement 1—source data 1.** Sequences used for phylogenetic tree, multi-alignments, MEME (Multiple Em for Motif Elicitation), and motif Weblogo in *Figure 2—figure supplement 1*.

**Figure supplement 1—source data 2.** Uncropped images of multi-alignments in *Figure 2—figure supplement 1C*.

**Figure supplement 1—source data 3.** MEME (Multiple Em for Motif Elicitation) analysis of *Fol*-Secreted Virulence-related Protein1 (FolSvp1) and its homologs sequences.

To confirm and extend these observations, FolSvp1R-GFP pulled down with anti-GFP agarose beads was analyzed by mass spectrometry following digestion with trypsin. In total, three ubiquitinated lysine residues, K152, K258, and K284, were identified (*Figure 3C*, *Figure 3—figure supplement 1A, B*). To examine the effect of ubiquitination on FolSvp1 degradation, we changed each of these three lysine residues, or all of them to R to eliminate ubiquitination of these sites in FolSvp1R-GFP. As shown

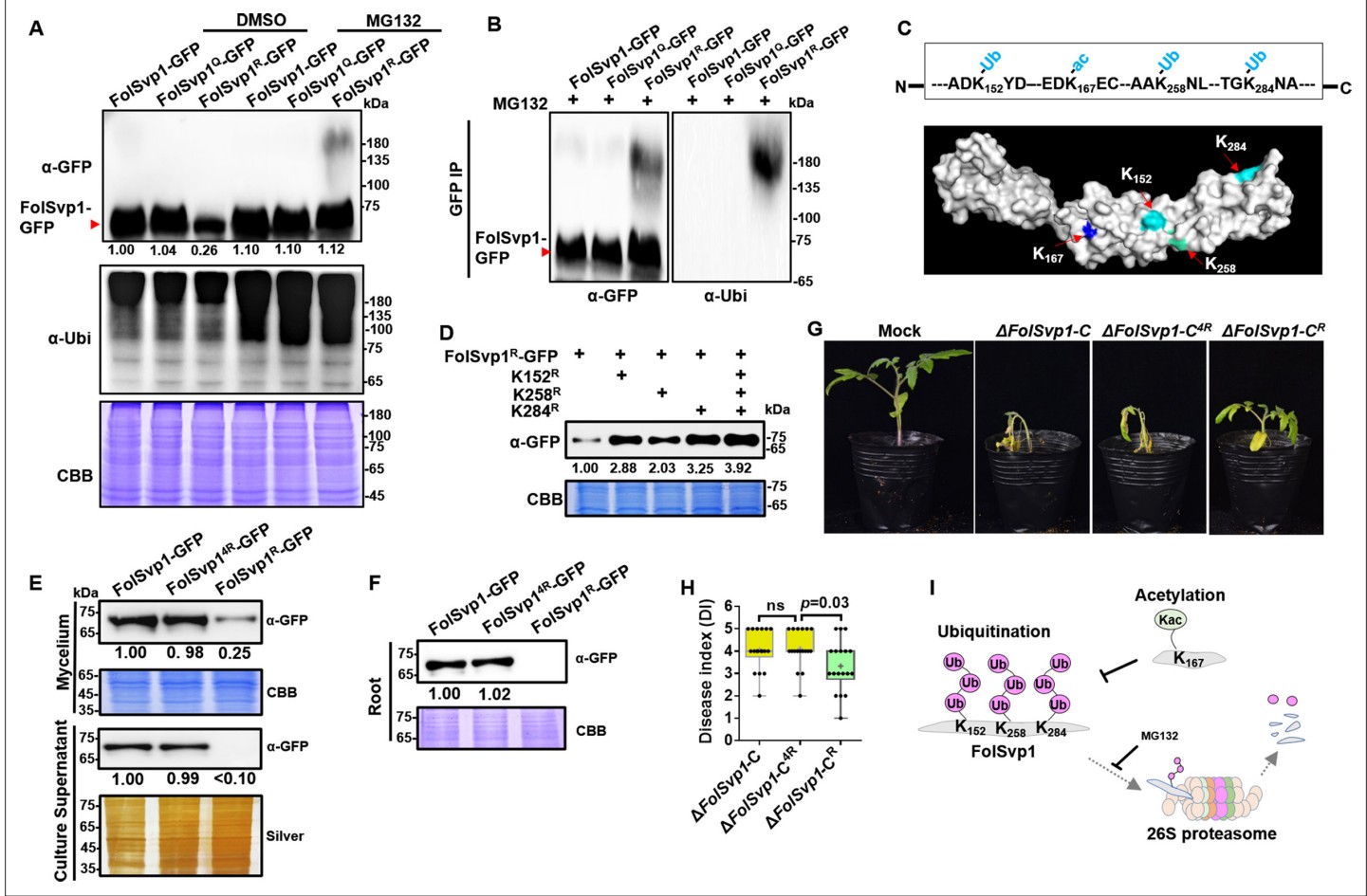

**Figure 3.** Amount and stability of wild-type (WT) and K167 mutant *Fol*-Secreted Virulence-related Protein1 (FolSvp1) proteins in *Fol*. (**A**) Amount of FolSvp1-GFP, FolSvp1$^Q$-GFP, and FolSvp1$^R$-GFP with or without MG132 treatment. Conidia of the indicated strains were cultured in 5% liquid YEPD for 14 hr, followed by treatment with DMSO or 50 µM MG132 for 2 hr. Total proteins extracted were probed with α-GFP or anti-ubiquitin (P4D1) monoclonal antibody (α-Ubi). The amount of WT FolSvp1 treated with DMSO was set as 1. (**B**) Ubiquitination of FolSvp1-GFP, FolSvp1$^Q$-GFP, and FolSvp1$^R$-GFP. FolSvp1 proteins pulled down from the indicated strains with GFP-Trap beads were probed with α-GFP or α-Ubi. (**C**) Ubiquitination of K152, K258, and K284 and acetylation of K167 (up), and their distribution within FolSvp1 according to the predicted protein modeling structure (bottom). (**D**) Amount of K152, K258, and K284 mutant FolSvp1$^R$ proteins. The amount of native FolSvp1$^R$ was set as 1. (**E**) Level of FolSvp1, FolSvp1$^R$ (K167R), and FolSvp1$^{4R}$ (simultaneous mutation of K152, K167, K258, and K284 to R) proteins in mycelia and their secreted amount. (**F**) Amount of the WT and mutant FolSvp1 proteins in tomato root cells. (**G**) Virulence of the WT and *FolSvp1* mutant strains on tomato. (**H**) Disease index scored at 14 days after inoculation (DAI). For (**E–H**), amount, secretion, virulence, and disease index were determined as in *Figure 2*. (**I**) Schematic representation of K167 acetylation inhibiting ubiquitination-dependent degradation of FolSvp1 in *Fol*. All the experiments were performed in the presence of tomato roots. Each gel shown is a representative experiment carried out at least twice.

The online version of this article includes the following source data and figure supplement(s) for figure 3:

**Source data 1.** Uncropped gels and blots in *Figure 3*.

**Source data 2.** Statistical analysis in *Figure 3*.

**Figure supplement 1.** Identification of the ubiquitinated sites in *Fol*-Secreted Virulence-related Protein1 (FolSvp1).

**Figure supplement 1—source data 1.** Liquid chromatography-tandem mass spectrometry (LC-MS/MS) identification of *Fol*-Secreted Virulence-related Protein1 (FolSvp1) ubiquitination in *Fusarium oxysporum* f. sp. *lycopersici* (*Fol*).

**Figure supplement 1—source data 2.** Liquid chromatography-tandem mass spectrometry (LC-MS/MS) identification of *Fol*-Secreted Virulence-related Protein1 (FolSvp1) ubiquitination in planta.

in *Figure 3D*, mutation of any of the three K to R markedly increased the amount of FolSvp1, while their simultaneous alteration elevated FolSvp1$^R$-GFP to the identical level of FolSvp1-GFP (*Figure 3D and E*). As a consequence, secretion of the 4R mutant FolSvp1 (FolSvp1$^{4R}$-GFP) was much increased (*Figure 3E*), and this protein could now be observed in the nucleus of infected root cells (*Figure 3F*).

In contrast to the impaired virulence of Δ*FolSvp1-C^R^*, complementation with *FolSvp1^4R^* completely restored the virulence phenotype of the Δ*FolSvp1* mutants (**Figure 3G and H**), further confirming the importance of FolSvp1 stability in *Fol* virulence. These results indicate first, that ubiquitination of K152, K258, and K284 leads to degradation of FolSvp1 by the 26S proteasome and second, that acetylation of K167 prevents ubiquitination-mediated FolSvp1 degradation, resulting in stabilization of this protein in *Fol* (**Figure 3I**).

## Acetylation of K167 by the lysine acetyltransferase FolArd1

To examine the relation between acetylation of K167 and FolSvp1 stability in more detail, it was of interest to identify the enzyme responsible for acetylation of this protein. For this purpose, we utilized the FolSvp1-GFP and the GFP transformants and performed immunoprecipitation (IP) followed by liquid chromatography-tandem mass spectrometry (LC-MS/MS) (**Supplementary file 1a and b**). Among the candidate-binding partners, we initially focused on FolArd1 (FOXG_05106), an ortholog

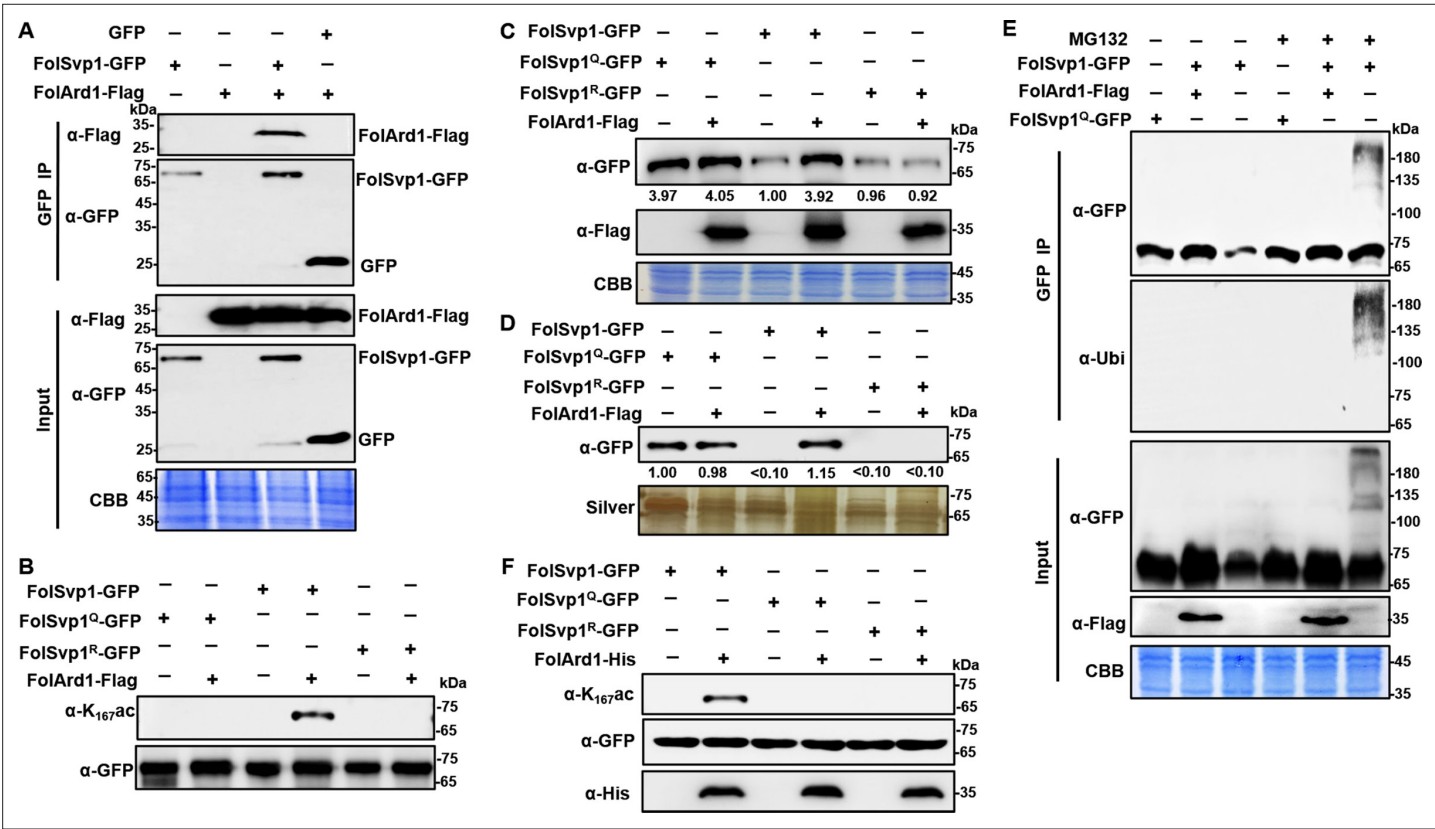

**Figure 4.** Acetylation of *Fol*-Secreted Virulence-related Protein1 (FolSvp1) by FolArd1. (**A**) Physical interaction of FolSvp1 and FolArd1 in vivo. Co-immunoprecipitation (co-IP) assays were performed as described in Materials and methods. Proteins pulled down with GFP-Trap beads were probed with α-GFP and α-Flag, respectively (top). Input proteins were shown by Western blotting with α-GFP and α-Flag, and CBB staining (bottom). (**B**) Acetylation of FolSvp1-GFP, FolSvp1^Q^-GFP, and FolSvp1^R^-GFP with or without *FolArd1* overexpression. Proteins pulled down with GFP-Trap beads were probed with α-GFP and α-K167ac, respectively. The same amount of FolSvp1-GFP, FolSvp1^Q^-GFP, and FolSvp1^R^-GFP was loaded to compare K167 acetylation of different samples. (**C**) Amount of FolSvp1-GFP, FolSvp1^Q^-GFP, and FolSvp1^R^-GFP with or without *FolArd1* overexpression. Input proteins were shown by Western blotting with α-GFP and α-Flag, and CBB staining. The amount of WT FolSvp1 without *FolArd1* overexpression was set as 1. (**D**) Amount of FolSvp1-GFP, FolSvp1^Q^-GFP, and FolSvp1^R^-GFP secreted with or without *FolArd1* overexpression. Proteins pulled down from culture supernatant with GFP-Trap beads were probed with α-GFP. The amount of secreted FolSvp1^Q^ without *FolArd1* overexpression was set as 1. (**E**) Ubiquitination of FolSvp1-GFP and FolSvp1^Q^-GFP with or without *FolArd1* overexpression, performed as in **Figure 3**. (**F**) FolArd1 directly acetylates FolSvp1 in vitro. FolSvp1-GFP, FolSvp1^Q^-GFP, or FolSvp1^R^-GFP pulled down with GFP-Trap beads were incubated with or without 5 μg of purified FolArd1 in the presence of 0.2 mM Ac-CoA. Products were then analyzed by immunoblotting using α-GFP, α-His, or α-K167ac. (**A**) was performed in the presence of tomato roots, and (**B–F**) were carried in the absence of tomato roots. Each gel shown is a representative experiment carried out at least twice.

The online version of this article includes the following source data for figure 4:

**Source data 1.** Uncropped gels and blots in **Figure 4**.

of the human lysine acetyltransferase ARD1 (*Lim et al., 2006*), which was significantly enriched in the FolSvp1-GFP sample.

FolSvp1-GFP and FolArd1-Flag fusion constructs were co-introduced into *F. oxysporum* protoplasts, and positive transformants were selected. FolArd1 was detected in the proteins that eluted from anti-GFP beads using the anti-Flag antibody (*Figure 4A*), suggesting that FolArd1 interacts with FolSvp1. To verify that FolArd1 plays a role in acetylation of FolSvp1 in vivo, we first tried to inactivate this enzyme. However, no deletion mutants were obtained after numerous attempts, suggesting that FolArd1 is likely an essential protein in *Fol*. Accordingly, we determined the effect of elevated FolArd1 on FolSvp1. Overexpression of *FolArd1* markedly increased acetylation of WT FolSvp1 in the absence of tomato roots, but had no obvious effect on the Q and R mutants (*Figure 4B*), indicating that FolArd1 is an enzyme that acetylates K167. Concomitantly, the accumulation of WT FolSvp1 in mycelia was elevated by ~4-fold (*Figure 4C*), leading to increased secretion of this protein (*Figure 4D*). Overexpression of *FolArd1* diminished FolSvp1 ubiquitination even after MG132 treatment (*Figure 4E*), indicating that acetylation efficiently prevents buildup of the transiently ubiquitinated species. Moreover, purified FolArd1 acetylated WT FolSvp1 in vitro, whereas it displayed no activity against either the K167Q or the K167R mutant FolSvp1 proteins (*Figure 4F*). These data demonstrate that FolArd1 is responsible for acetylation of K167 in FolSvp1, and they also show that acetylation directly regulates the accumulation of FolSvp1 in fungal cells and its secretion.

## Acetylation stabilizes FolSvp1 in planta

FolSvp1 contains an SP that facilitates its secretion out of the fungal cells (*Figure 1A*, *Figure 2—figure supplement 1B*). As in tomato roots, FolSvp1 was localized in the nuclear but not the apoplastic regions of *Nicotiana benthamiana* leaves in either the absence or presence of the SP (*Figure 5A*), indicating that the native SP of FolSvp1 ineffectively promotes its secretion in planta (*Petre and Kamoun, 2014*). Prediction with the online tool cNLS Mapper revealed two potential importin α-dependent nuclear localization signals (NLSs) in FolSvp1, namely NLS1 (aa 94–125) and NLS2 (aa 281–310) (*Figure 5B*). To determine how FolSvp1 is translocated into the nucleus, we replaced the core hydrophilic residues K, R, and Q with the hydrophobic residue isoleucine (*Kosugi et al., 2009a*, *Rowland et al., 2003*). As shown in *Figure 5B*, NLS1 mutation led to impaired localization of FolSvp1, while NLS2 mutation had no obvious effect on its nuclear distribution. Consequently, mutation of NLS1 (Δ*FolSvp1-C^{nls1}*) but not NLS2 (Δ*FolSvp1-C^{nls2}*) led to reduced virulence of *Fol* (*Figure 5C and D*). These results suggest that NLS1 dominates the nuclear translocation of FolSvp1 required for pathogenicity of this fungus.

The SP of PR1 is highly effective in targeting proteins/peptides to the apoplast via the ER/Golgi secretory pathway (*Kloppholz et al., 2011*). To manipulate FolSvp1 secretion in tobacco, we replaced its native SP with that of tomato SlPR1 and found that FolSvp1 localization was dramatically changed, with clear observation of the protein in the apoplast (*Figure 5A and E*). To explore the function of K167 acetylation in planta, we replaced it with Q or R, and found this modification has no effect on the distribution of FolSvp1 with or without SPs (*Figure 5A*). Interestingly, although there is little difference among different forms of nuclear FolSvp1 (*Figure 5F–I*), the amount of the Q and R mutants was ~3-fold higher than the WT FolSvp1 distributed in the cytoplasm (*Figure 5H and I*). As the amount of the Q and R mutants are almost equal, we speculate that K167 itself is a target of the plant cytoplasmic ubiquitination system. Not unexpected, mass spectrometry analysis with 76.9% coverage of FolSvp1 (*Figure 3—figure supplement 1C*) identified only one ubiquitinated residue, K167 (*Figure 3—figure supplement 1D*). MG132 treatment led to increased amount and detection of the ubiquitinated form with only the WT FolSvp1, but not the Q and R mutants (*Figure 5J and K*), suggesting that acetylation blocks K167 ubiquitination-dependent degradation of FolSvp1 before entering the host nucleus. Collectively, these data indicate that K167 acetylation is required for FolSvp1 stability but not its translocation into host nucleus in planta.

## FolSvp1 targets tomato SlPR1

To gain further insight into the function of FolSvp1 in the host, we used FolSvp1 without the SP (^{ΔSP}FolSvp1) as a bait to screen the tomato cDNA library via GAL4-based yeast two-hybrid (Y2H) system and identified 13 FolSvp1-interacting protein candidates (*Figure 6—figure supplement 1A*). Among them, two genes were identified as frameshift, nine genes were not full length, and one mitochondrial outer membrane protein (SlMit) had self-activation activity (*Figure 6—figure supplement*

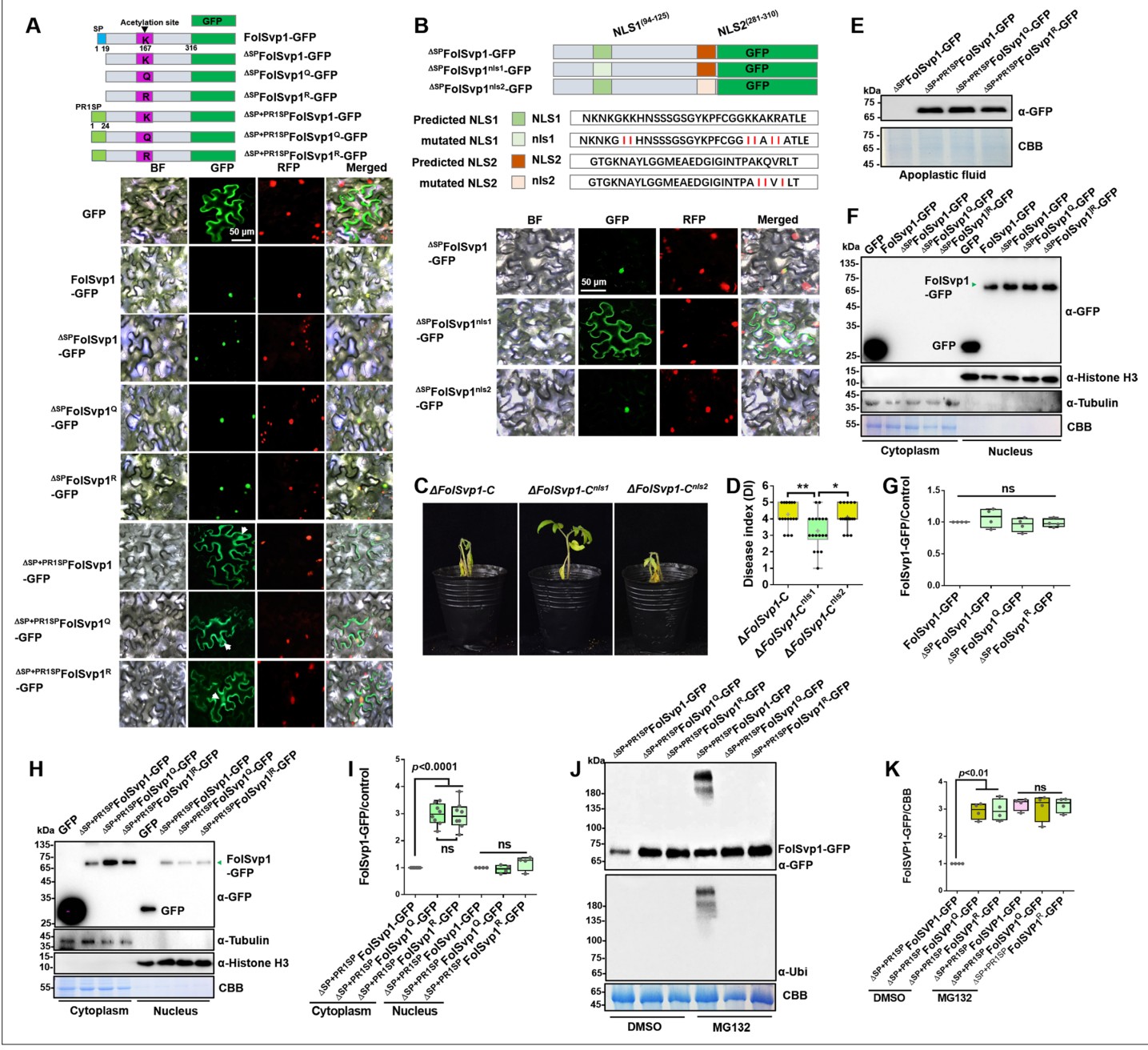

**Figure 5.** Subcellular localization and stability of *Fol*-Secreted Virulence-related Protein1 (FolSvp1) in planta. (**A**) Subcellular localization of wild-type (WT) and K167 mutant FolSvp1-GFP proteins in tobacco leaves. ΔSP indicates removal of the native signal peptide of FolSvp1, and ΔSP + PR1SP indicates replacement of its native signal peptide with the plant PR1 signal peptide. (**B**) Subcellular localization of FolSvp1-GFP proteins with the native or mutant nuclear localization signal (NLS) in tobacco leaves. For (**A**) and (**B**), the indicated constructs (top) were transiently expressed in Histone H2B-RFP overexpression *Nicotiana benthamiana*, and images were taken at 3 days after inoculation (DAI) (bottom). Scale bars = 50 µm. (**C**) Virulence of the indicated strains on tomato. (**D**) Disease index scored at 14 DAI. For (**C–D**), virulence and disease index were determined as in *Figure 2*. (**E**) Amount of WT and K167 mutant FolSvp1 proteins in the apoplast of tobacco leaves. Proteins were extracted from apoplast as described in Materials and methods and probed with α-GFP. Coomassie brilliant blue (CBB) staining shows equal protein loading to each lane. (**F**) Subcellular fractionation of WT and K167 mutant FolSvp1-GFP proteins with or without its native signal peptide in tobacco leaves. Nuclear and cytoplasmic proteins were separately extracted and FolSvp1-GFP were detected with α-GFP. The fractionation controls were tubulin (cytoplasm), histone H3 (nucleus), and CBB staining. (**G**) Quantification of nuclear FolSvp1-GFP proteins relative to histone H3 in (**F**). Statistical significance was revealed by one-way analysis of variance (ANOVA) (mean ± SE of four independent biological replicates, p-values are shown). (**H**) Subcellular fractionation of WT and K167 mutant ΔSP+PR1SPFolSvp1-GFP proteins in tobacco leaves. (**I**) Quantification of cytoplasmic and nuclear FolSvp1-GFP proteins relative to tubulin and histone H3, respectively, in (**H**). Statistical significance was revealed by one-way ANOVA (mean ± SE of at least four independent biological replicates, n=8 for

*Figure 5 continued on next page*

*Figure 5 continued*

cytoplasm, n=4 for nucleus, p-values are shown). (**J**) Amount and ubiquitination of WT and K167 mutant ᐃSP+PR1SPFolSvp1-GFP proteins in the cytoplasm of tobacco leaves with or without MG132 treatment. Three days after overexpression of the indicated constructs, tobacco leaves were treated with DMSO or MG132 for 4 hr and cytoplasmic proteins were extracted. Proteins pulled down with GFP-Trap beads were probed with α-GFP and α-Ubi, respectively. CBB staining shows protein loading to each lane. (**K**) Quantification of ᐃSP+PR1SPFolSvp1-GFP proteins relative to CBB staining in (**J**). Statistical significance was revealed by p-value from one-way ANOVA analysis (mean ± SE of four independent biological replicates, p-values are shown).

The online version of this article includes the following source data and figure supplement(s) for figure 5:

**Source data 1.** Uncropped gels and blots in *Figure 5*.

**Source data 2.** Statistical analysis in *Figure 2*.

**Figure supplement 1.** Subcellular localization of SlPR1 with or without its N-terminal signal peptide in tobacco leaves.

**Figure supplement 1—source data 1.** Uncropped gels and blots in *Figure 5—figure supplement 1*.

*1B*). Finally, SlPR1 (also named as PR1b1/P14a/P6 in tomato; Genbank accession number P04284) was selected for further research. qRT-PCR analysis detected the transcript accumulation of *SlPR1* in multiple tissues including cotyledon, root, leaves, flowers, green fruit, and red fruit (*Figure 6A*). In addition, the expression of *SlPR1* in the root was up-regulated after *Fol* infection, reaching a peak at 3 days after inoculation (DAI) (*Figure 6B*). These results suggest a role of SlPR1 in tomato immunity and thus a likely target to be attacked by *Fol*.

To verify the interaction between FolSvp1 and SlPR1, an Y2H assay was performed for the mature form of the proteins without the SP element (*Breen et al., 2016*; *Sung et al., 2021*). As shown in *Figure 6C*, ᐃSPFolSvp1 and ᐃSPSlPR1 indeed associate. To determine whether FolSvp1 is able to interact with SlPR1 localized to the apoplast, its original SP was replaced with the SP of SlPR1 and an in vivo bimolecular fluorescence complementarity (BiFC) assay was carried out. The observation of yellow fluorescence indicated that FolSvp1 secreted by the fungus can associate with SlPR1 in the extracellular space (*Figure 6D*). In either the in vitro or the in vivo experiments, change of K167 to Q or R has little effect on the interaction of FolSvp1 with SlPR1 (*Figure 6C and D*).

Besides PR1b1, tomato contains 12 genes that may encode PR1 proteins. Among them, 10 genes were up-regulated after *Fol* infection (*Figure 6—figure supplement 2A*) and were then tested for the interaction with FolSvp1. Both Y2H and BiFC analyses indicated that FolSvp1 only associates with SlPR1 (PR1b1) but not other PR1 proteins (*Figure 6—figure supplement 2B, C*). Based on these data, we propose that FolSvp1 specifically targets SlPR1 in planta independent of its acetylation.

## SlPR1 contributes to tomato resistance

SlPR1 is a 159-amino acid protein with a putative SP on the N-terminus (aa 1–24) (*Figure 5—figure supplement 1A*), a key feature for secretion. To determine its subcellular localization, SlPR1 with or without the SP (ᐃSPSlPR1) was fused to RFP and transiently expressed in *N. benthamiana* (*Figure 5—figure supplement 1B*). After plasmolysis, SlPR1 was found to be located to the apoplast while ᐃSPSlPR1 was distributed in the cytoplasm (*Figure 5—figure supplement 1C*). These results confirmed the secretion of SlPR1 dependent on the N-terminal SP, which was accordingly used for manipulating localization of FolSvp1 in plant cells (*Figure 5A*).

To determine the role of SlPR1 in tomato resistance to *Fol* infection, we tried to delete *SlPR1* by CRISPR-cas9. However, after numerous efforts we could not obtain the desired knockout mutants, suggesting that such mutations of *SlPR1* in tomato are lethal. Alternatively, we generated *SlPR1* overexpression plants (*Figure 6E*) and carried out *Fol* inoculation assays. Compared with the WT plants, the *SlPR1* overexpression seedlings (SlPR1(OE)) showed significantly enhanced resistance to *Fol* (*Figure 6F and G*). Similar to AtCAPE1 (*Chien et al., 2015*), the CAPE1-GFP band with the molecular weight of 29.8 kDa was observed after *Fol* infection (*Figure 6E*). Mass spectrometry analysis of the band revealed the presence of only the CAPE1 region but not other parts of SlPR1, further confirming it to be CAPE1-GFP. In contrast, no such band was observed in seedlings overexpressing SlPR1 without the C-terminal 15-amino acid CAPE1 (ᐃCAPE1SlPR1(OE)) (*Figure 6E*) which exhibited unchanged resistance to *Fol* (*Figure 6F and G*). To further confirm this peptide contributes to plant resistance, tomato seedlings treated with or without synthetic CAPE1 were inoculated with *Fol*. As shown in *Figure 6H and I*, pretreatment with CAPE1 effectively suppressed *Fol* invasion. These results

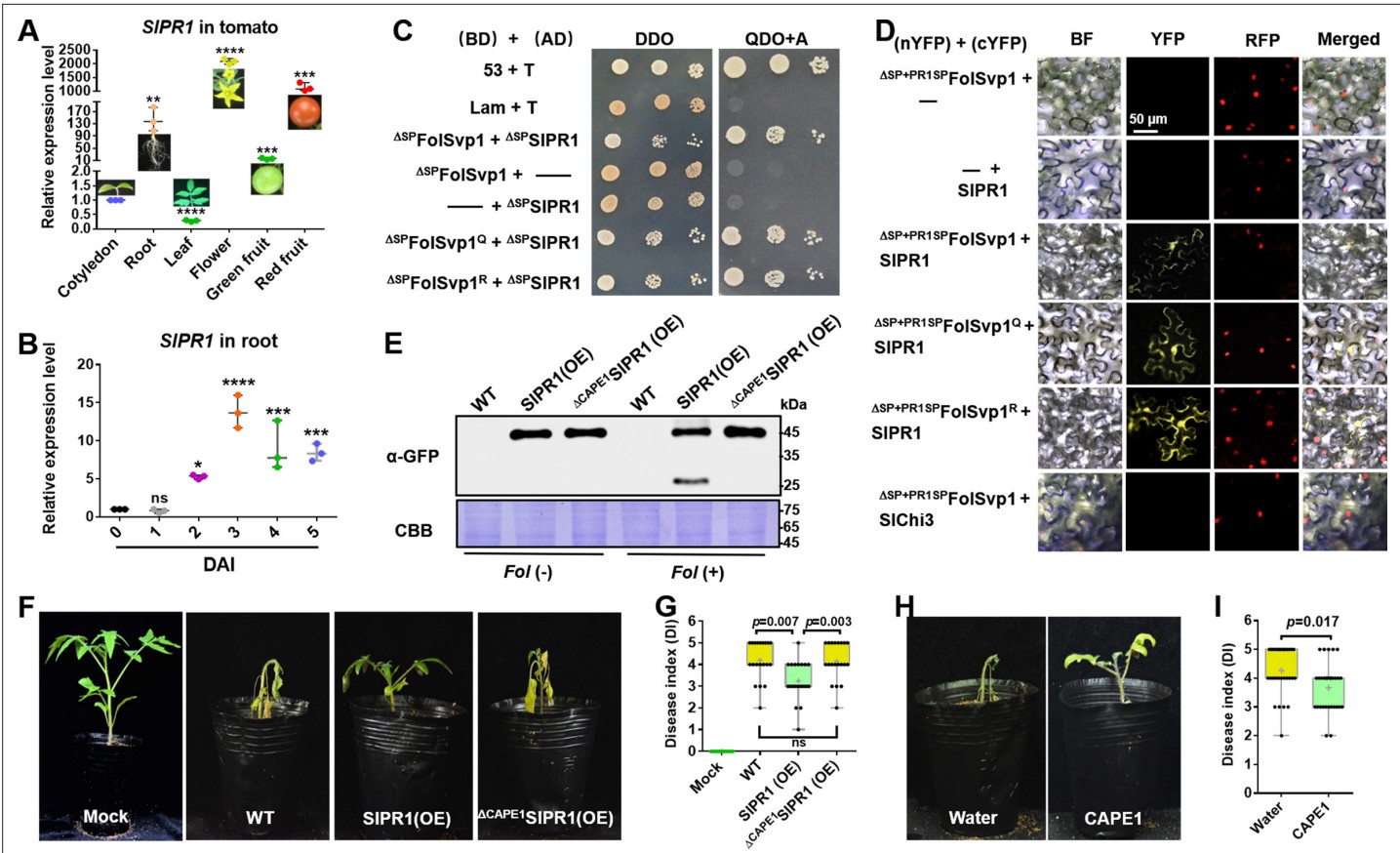

**Figure 6.** Physical interaction of *Fol*-Secreted Virulence-related Protein1 (FolSvp1) with SlPR1 and contribution of SlPR1 to tomato resistance. (**A**) Expression profile of *SlPR1* in cotyledon, root, leaves, flowers, green fruit, and red fruit measured by qRT-PCR. (**B**) Expression profile of *SlPR1* in tomato roots after *Fusarium oxysporum* f. sp. *lycopersici* (*Fol*) inoculation at the indicated times measured by qRT-PCR. For (**A**) and (**B**), the expression levels were normalized to that of the *SlActin*. Data represent means ± SE of three independent replicates. Statistical comparisons (Student's t-test) were presented for each variable (*p<0.05, **p<0.01, ***p<0.001, ****p<0.0001). (**C**) Yeast two-hybrid (Y2H) assays showing direct interaction of SlPR1 with wild-type (WT) and K167 mutant FolSvp1 proteins in vitro. T was co-transformed with p53 or Lam to represent the positive and the negative control, respectively. ΔSP indicates removal of the native signal peptide. Serial dilutions from cell suspensions of a single yeast colony were shown to represent the strength of interaction. Images were taken at 3 days after inoculation (DAI). (**D**) Bimolecular fluorescence complementarity (BiFC) assays showing interaction of SlPR1 with WT and K167 mutant ΔSP+PR1SP FolSvp1 proteins in vivo. The indicated constructs were transiently expressed in Histone H2B-RFP overexpression *Nicotiana benthamiana*, and images were taken at 3 DAI. cYFP, C-terminal region of YFP; nYFP, N-terminal region of YFP. Scale bars = 50 μm. (**E**) Western blot analysis showing the expression of SlPR1-GFP proteins and the generation of CAPE1-GFP with (+) or without (-) *Fol* infection in tomato. Total proteins extracted were probed with α-GFP. Coomassie brilliant blue (CBB) staining shows protein loading to each lane. (**F**) Resistance of WT and *SlPR1* overexpression tomato seedlings to *Fol* infection. (**G**) Disease index scored at 14 DAI of 20 plants in (**F**). Statistical significance was revealed by Student's t-test. p-Values are shown. (**H**) Resistance of WT tomato seedlings treated with or without CAPE1 to *Fol* infection. Inoculation was carried out 6 hr after application of 250 nM synthetic CAPE1 peptide to tomato roots. (**I**) Disease index scored at 14 DAI of 20 plants in (**H**). Statistical significance was revealed by Student's t-test. p-Values are shown.

The online version of this article includes the following source data and figure supplement(s) for figure 6:

**Source data 1.** Uncropped gels and blots in *Figure 6*.

**Source data 2.** Statistical analysis in *Figure 6*.

**Figure supplement 1.** Identification of *Fol*-Secreted Virulence-related Protein1 (FolSvp1) interacting proteins in tomato.

**Figure supplement 2.** Expression profile and interaction of tomato PR1 proteins with *Fol*-Secreted Virulence-related Protein1 (FolSvp1).

**Figure supplement 2—source data 1.** qRT-PCR dataset of 13 genes used for *Figure 6—figure supplement 2A*.

indicate that SlPR1 is a critical part of tomato immunity against *Fol* through generation of the CAPE1 peptide.

## FolSvp1 relocates SlPR1 to host nucleus to interfere with CAPE1 generation

SlPR1 is an apoplast-localized protein (*Figure 5—figure supplement 1*), but BiFC analyses indicate its association with FolSvp1 in the nucleus (*Figure 6D*). One possible explanation is that FolSvp1 can change the subcellular localization of SlPR1 by direct interaction. To determine this possibility, we incubated tomato root protoplasts with purified MBP-SlPR1 and found that the addition of FolSvp1-GFP led to detection of MBP-SlPR1 in the nucleus (*Figure 7A*). In contrast, FolSvp1 did not interact with the secreted chitinase (*Aimé et al., 2013*), SlChi3 (*Figure 6D*), and failed to carry it into the nucleus (*Figure 7A*), further confirming the specificity of FolSvp1 on SlPR1. Although the acetylation status of K167 has no effect on the function of FolSvp1, mutation of NLS1 (FolSvp1$^{nls1}$-GFP) abolished the relocation of SlPR1 to the nucleus (*Figure 7B*). Moreover, SlPR1 was observed in the nucleus of *SlPR1* overexpression plants infected by WT but not the Δ*FolSvp1* and the Δ*FolSvp1-C$^{nls1}$* mutant strains (*Figure 7C*). The observations that FolSvp1 associates with the N-terminal part of SlPR1 but not the CAPE1 region (*Figure 7—figure supplement 1*) and that no detection of CAPE1-GFP in the nucleus of *Fol*-infected root cells (*Figure 7C*) exclude the possibility that FolSvp1 relocates SlPR1 released CAPE1 during plant immunity. Collectively, these results suggest that FolSvp1 hijacks apoplastic SlPR1 and translocates it into the host nucleus dependent on the NLS1 motif.

To determine the effect of SlPR1 relocation, we replaced its native SP with an NLS and overexpressed the protein in tomato. As shown in *Figure 7D and E*, substitution NLS for SP led to the localization of SlPR1 to the nucleus. Therefore, NLS-$^{ΔSP}$SlPR1-GFP can be used to mimic FolSvp1-mediated SlPR1 nuclear relocation. In contrast to SlPR1, no CAPE1 was produced in NLS-$^{ΔSP}$SlPR1-GFP overexpression plants after *Fol* invasion (*Figure 7F*) and no increased resistance was observed (*Figure 7G and H*). To further confirm the relation between FolSvp1 triggered SlPR1 translocation and CAPE1 generation, we inoculated the *SlPR1* overexpression seedlings with the WT and *FolSvp1* overexpression strains. Compared with the WT strain, much less CAPE1 was produced after inoculation with the *FolSvp1* overexpression strain (*Figure 7I*), and as a consequence, a more severe phenotype was observed (*Figure 7J and K*). These results indicate that relocation of SlPR1 into the host nucleus by FolSvp1 eliminates CAPE1 generation and abolishes its contribution to plant resistance.

Based on the above data, we propose a model to explain how FolSvp1 manipulates plant immunity (*Figure 8*). In response to *Fol* infection, tomato up-regulates the expression level and secretion of the PR protein SlPR1 to the apoplast to enhance resistance through generation of the CAPE1 peptide. However, *Fol* produces the effector FolSvp1 to counter the SlPR1-dependent defense response. FolSvp1 directly interacts with and hijacks the secreted SlPR1, relocating it into the host nucleus, thereby eliminating CAPE1 production and enabling *Fol* to invade tomato successfully. Critically, K167 acetylation of FolSvp1 by FoArd1 acts as a protective shield to prevent ubiquitination-dependent degradation of FolSvp1 in both *Fol* and plant cells, allowing it to function properly in *Fol* invasion.

## Discussion

Although effectors are critical virulence factors of plant pathogens, relatively little is known about how these proteins might be regulated. The studies described here provide evidence that lysine acetylation can regulate the stability of an important fungal effector that specifically interacts with and inactivates a key host defense response protein, thereby enabling virulence of the pathogen. Specifically, we found that (i) FolSvp1, a previously uncharacterized effector, is required for the full virulence of *Fol*; (ii) tomato SlPR1 is capable of producing the bioactive peptide CAPE1 after *Fol* invasion and represents a crucial contributor to host defense response; (iii) the secreted FolSvp1 eliminates CAPE1 generation through hijacking the tomato apoplastic protein SlPR1 to host nucleus to promote fungal invasion; (iv) addition of an acetyl group on K167 by FolArd1 stabilizes FolSvp1 in both *Fol* and plant cells; and (v) acetylation of FolSvp1 directly regulates *Fol* virulence. As such, these findings greatly expand our understanding of effector regulation and open up new possibilities for investigation in the field of microbe-plant interactions. This complex regulatory system serves as a detailed example of how pathogens manipulate host immunity through regulation of an effector.

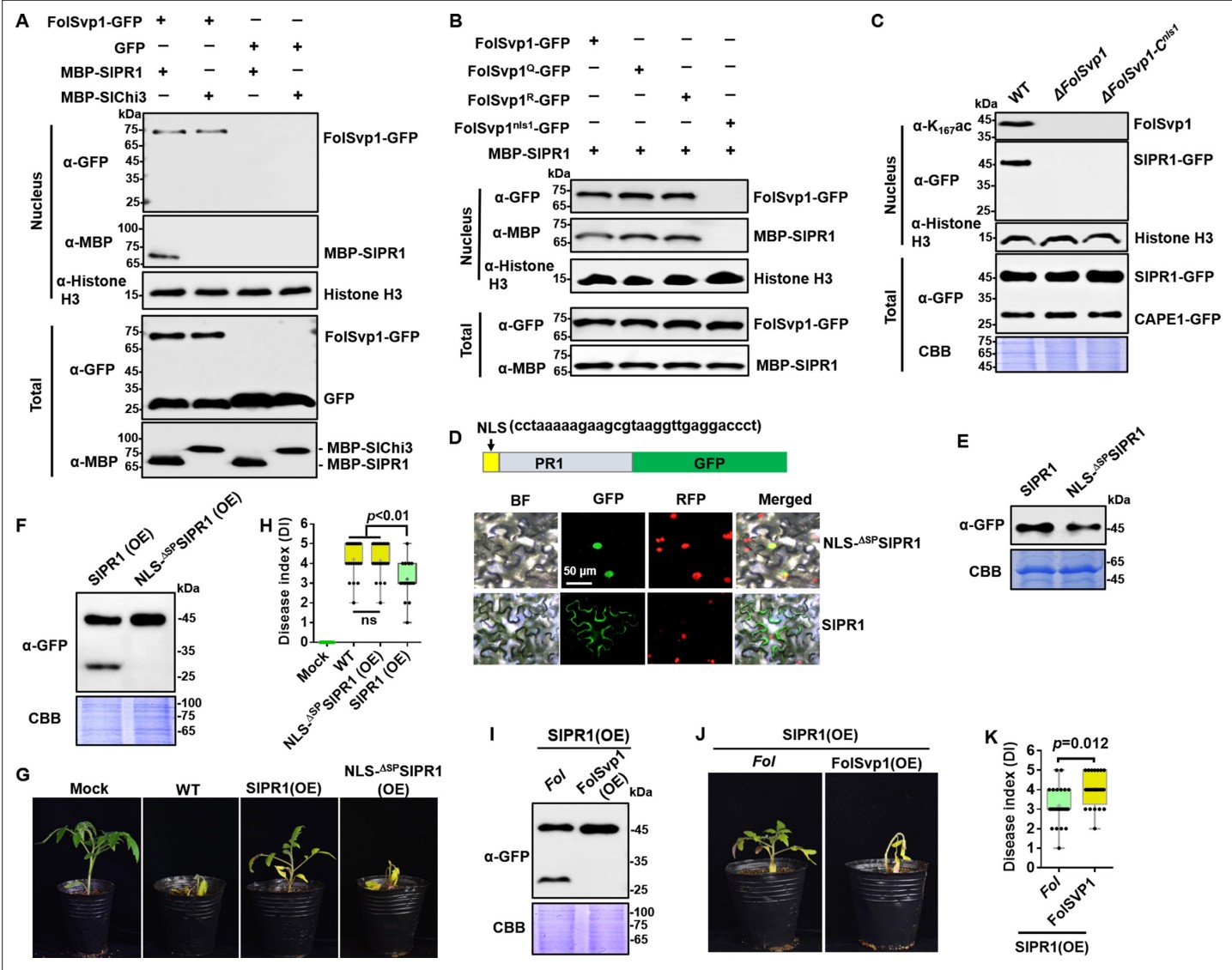

**Figure 7.** Relocation of SlPR1 into nucleus by *Fol*-Secreted Virulence-related Protein1 (FolSvp1) suppresses tomato resistance. (**A**) Import of SlPR1 into the nucleus of tomato root protoplast by FolSvp1. (**B**) Import of SlPR1 into the nucleus of tomato root protoplast by wild-type (WT) and mutant FolSvp1 proteins. For (**A**) and (**B**), 50 µg purified SlPR1-MBP or SlChi3-MBP was incubated with protoplast prepared from tomato root in the presence of GFP or FolSvp1-GFP proteins pulled down from *Fusarium oxysporum* f. sp. *lycopersici* (*Fol*). Four hours after incubation, nuclear proteins extracted were probed with α-GFP, α-MBP, and α-histone H3, respectively (top). Input proteins were shown by Western blotting with α-GFP and α-MBP (bottom). (**C**) Detection of SlPR1 in the nucleus of *SlPR1* overexpression tomato root infected by WT or *FolSvp1* mutant strains. Nuclear proteins extracted were probed with α-K$_{167}$ac, α-GFP, and α-histone H3, respectively (top). Input proteins were shown by Western blotting with α-GFP and Coomassie brilliant blue (CBB) staining (bottom). (**D**) Subcellular localization of SlPR1-GFP and NLS-$^{\Delta SP}$SlPR1-GFP in tobacco leaves. The DNA sequence for the nuclear localization signal (NLS) used here is shown in the inset. The indicated constructs were transiently expressed in *Nicotiana benthamiana*, and images were taken at 3 days after inoculation (DAI). Scale bars = 50 µm. (**E**) Western blot analysis showing the expression of SlPR1-GFP and NLS-$^{\Delta SP}$SlPR1-GFP in (**D**). Total proteins extracted were probed with α-GFP. CBB staining shows protein loading to each lane. (**F**) Western blot analysis showing the amount of SlPR1-GFP and CAPE1-GFP in SlPR1 and NLS-$^{\Delta SP}$SlPR1 overexpression tomato seedlings after *Fol* infection. (**G**) Resistance of WT and SlPR1 or NLS-$^{\Delta SP}$SlPR1 overexpression tomato seedlings to *Fol* infection. (**H**) Disease index scored at 14 DAI of 20 plants in (**G**). Statistical significance was revealed by one-way analysis of variance (ANOVA). p-Values are shown. (**I**) Western blot analysis showing the amount of SlPR1-GFP and CAPE1-GFP in SlPR1 overexpression tomato seedlings after infection by WT (*Fol*) or *FolSvp1* overexpression (FolSvp1(OE)) strains. (**J**) Resistance of SlPR1 overexpression tomato seedlings to WT and *FolSvp1* overexpression strains. (**K**) Disease index scored at 14 DAI of 24 plants in (**J**). Statistical significance was revealed by one-way ANOVA. p-Values are shown. Each gel shown is a representative experiment carried out at least twice.

The online version of this article includes the following source data and figure supplement(s) for figure 7:

**Source data 1.** Uncropped gels and blots in *Figure 7*.

*Figure 7 continued on next page*

*Figure 7 continued*

**Source data 2.** Statistical analysis in *Figure 7*.

**Figure supplement 1.** Association regions of SlPR1 with *Fol*-Secreted Virulence-related Protein1 (FolSvp1).

The cross-talk between acetylation and ubiquitination, which both happen at lysine residues, is a critical regulatory mechanism controlling vital cellular functions (*Caron et al., 2005*). The ubiquitination/26S proteasome mechanism is indispensable for the turnover of unuseful and foreign invading toxic proteins (*Yin, 2021*). As FolSvp1 targets SlPR1 to promote *Fol* invasion, it is not surprising that plants might evolve a defense by triggering degradation of FolSvp1 by ubiquitination. As a counteraction, acetylation of K167 would have evolved to prevent the degradation of FolSvp1 in plants. The detection of K167 ubiquitination and the identical behavior of the Q and R mutants in plant cells indicates a direct competition between acetylation and ubiquitination. Upon sensing of tomato roots by *Fol*, K167 is acetylated to stabilize FolSvp1, which would otherwise be turned over. The observations of ubiquitination of K152, K258, and K284, but not K167, together with the completely opposite behavior of the K167Q and K167R mutants, point to an indirect mechanism between these two modification types. Therefore, acetylation of a single lysine residue, K167, facilitates FolSvp1 secretion and

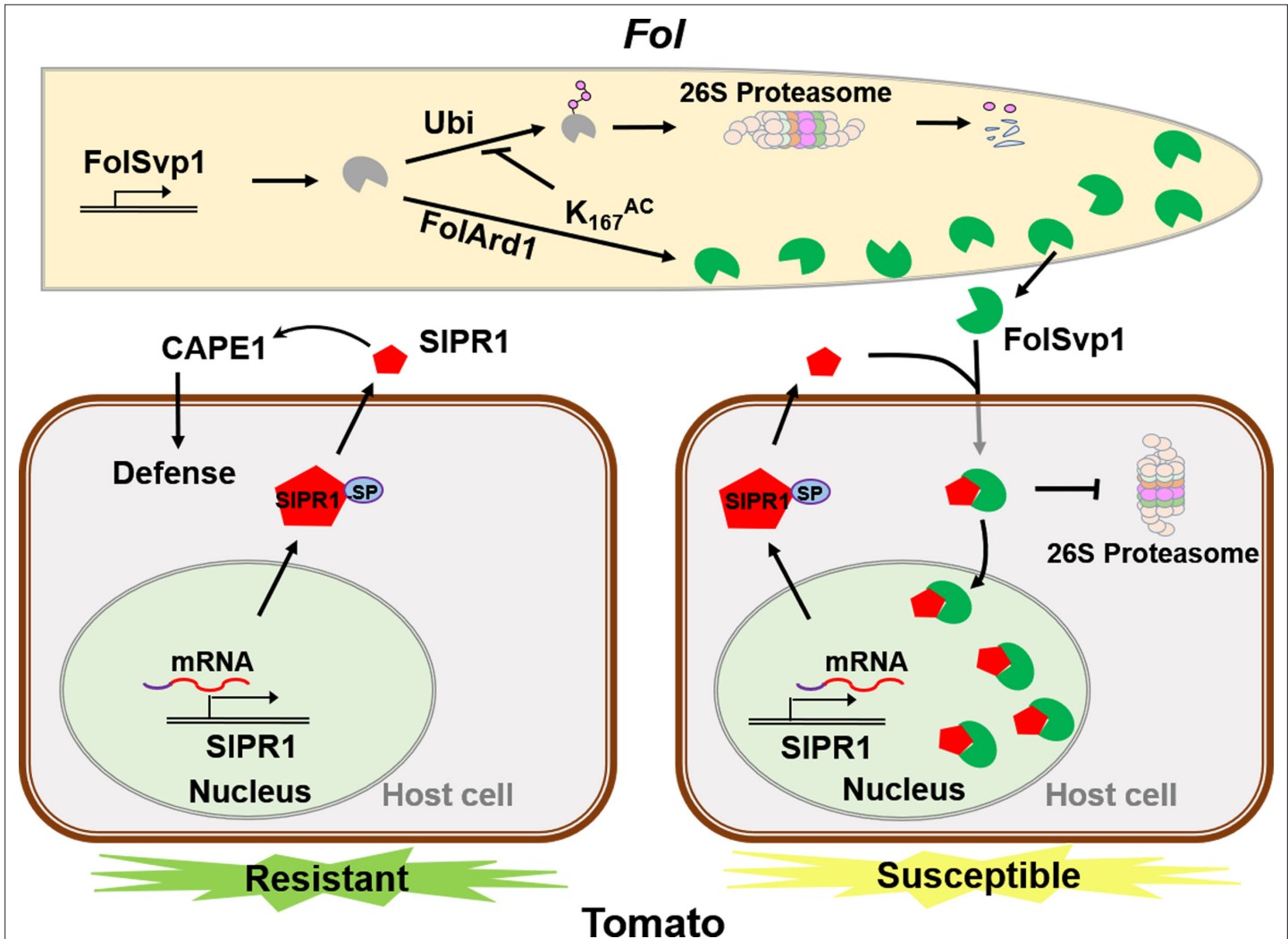

**Figure 8.** A model to explain the function of *Fol*-Secreted Virulence-related Protein1 (FolSvp1) in suppression of plant resistance. During *Fusarium oxysporum* f. sp. *lycopersici* (*Fol*) invasion, synthesized FolSvp1 is acetylated on K167 by FolArd1. This modification represses ubiquitination-dependent degradation of FolSvp1, resulting in its secretion. K167 acetylation also prevents the degradation of FolSvp1 by the plant 26S proteasome, enabling it to translocate SlPR1 from the apoplast into plant nucleus. As a consequence, CAPE1 production is abolished, and *Fol* invades successfully.

subsequently its virulence in planta with different mechanisms. Notably, K167 and its acetylated form, Q, are widely distributed in FolSvp1 homologs in a variety of plant pathogens, strongly suggesting that this modification is a conserved phenomenon determining proteins' stability during pathogen-plant interactions. The mechanism of acetylation-dependent stabilization of FolSvp1 in *Fol* is a topic to be explored in more detail in future studies, but it is clear that these results serve as a salient sample of how lysine acetylation regulates a protein's stability by cross-talk with ubiquitination.

PR1 is generally considered to be associated with defense against biotrophs or hemi-biotrophs (*Glazebrook, 2005*), however, its role in plant defense to *Fol*, one of the facultative parasitic fungi (*Divon et al., 2005*), is poorly characterized. The data described here provide evidence that SlPR1 contributes to host resistance to *Fol* through generating the defense signaling CAPE1 peptide. Now, it is known that the CAPE1 peptide demonstrates broad activity against *Parastagonospora nodorum* and *Pseudomonas syringae* DC3000 through activating defense-related genes (*Breen et al., 2017*; *Chen et al., 2014*; *Sung et al., 2021*). Moreover, PR1 proteins are also able to inhibit the growth of oomycetes (*Choudhary and Schneiter, 2012*; *Gamir et al., 2016*, *Niderman et al., 1995*; *Woloshuk et al., 1991*). Given its importance in plant defense responses, plant pathogens must eliminate the effect of PR1 to ensure invasion via multiple strategies. In fact, PR1 is targeted by diverse pathogen-secreted effectors, including CSEPP0055, ToxA, and SsCP1 with different mechanisms (*Breen et al., 2017*), and *P. nodorum* Tox3 was recently reported to prevent CAPE1 release from wheat PR1 in vitro (*Sung et al., 2021*). We show here that FolSvp1 physically interacts with tomato SlPR1 and translocates it from the apoplast to the host nucleus to block CAPE1 production, a phenomenon not previously reported in other pathogens. All these findings indicate that plant pathogens employ a variety of approaches to eliminate the function of PR1 proteins. The observations of FolSvp1 and SlPR1 interaction in both the apoplast and the nucleus of *N. benthamiana* leaves suggest that binding of FolSvp1 to SlPR1 may inhibit its anti-fungal activity and/or the cleavage of SlPR1 to produce CAPE1 in the extracellular region or even the cytoplasm. In addition, the BiFC assays performed with *N. benthamiana* leaves might not completely mimic the physiological conditions. Therefore, whether FolSpv1-mediated translocation of SlPR1 into the nucleus impedes CAPE1 release is the only way of PR1 inactivation needs to be further strengthened with additional data including BiFC analyses using the *FolSvp1* overexpression *Fol* strain and the *SlPR1* overexpression tomato in future studies.

These studies also provide a clear example in which acetylation affects the stability of an effector in plant pathogens. Acetylation has been shown to affect many proteins and processes in both prokaryotic and eukaryotic systems, but to date, only a few instances of a specific effect on a virulence factor have been uncovered. The discovery here that acetylation of one specific lysine residue in FolSvp1 increases the stability of this protein in *Fol* and in planta, promoting invasion of tomato, provides a clear example of a direct effect of acetylation on a biological process in plant pathogens. With the recent discovery that a large number of putative effectors are acetylated in fungi (*Li et al., 2020a*, *Lv et al., 2016*; *Zhou et al., 2016*), *Phytophthora* (*Li et al., 2016*), and bacteria (*Li et al., 2020b*, *Ren et al., 2017*), it is likely that the findings reported here are only the beginning of what will be a widespread phenomenon in plant pathogens.

# Materials and methods

**Key resources table**

| Reagent type (species) or resource | Designation | Source or reference | Identifiers | Additional information |
|---|---|---|---|---|
| Gene (*Fusarium oxysporum* f. sp. *lycopersic*) | *FolSvp1* | *Fusarium* genome resource | XP_018249666 | |
| Gene (*Fusarium oxysporum* f. sp. *lycopersic*) | *FolArd1* | *Fusarium* genome resource | XP_018240107 | |
| Gene (*Solanum lycopersicum*) | *SlPR1* | *Solanum* genome resource | P04284 | |
| Gene (*Solanum lycopersicum*) | *SlChi3* | *Solanum* genome resource | Z15141 | |
| Genetic reagent (*Nicotiana benthamiana*) | RFP:H2B | *Martin et al., 2009* | | |
| Strain, strain background (*Escherichia coli*) | *E. coli* BL21 (DE3) | Tsingke Biotechnology | Cat. #: TSC-E05 | |
| Strain, strain background(*Agrobacterium tumefaciens*) | LBA4404 | AngYuBio | Cat. #: AYBIO-G6038 | |

*Continued on next page*

*Continued*

| Reagent type (species) or resource | Designation | Source or reference | Identifiers | Additional information |
|---|---|---|---|---|
| Strain, strain background (Yeast) | Y2HGold | 2nd Lab | Cat. #: YC1002 | |
| Antibody | Anti-GFP (mouse monoclonal) | Abcam | Cat. #: ab183734 | 1:5000 |
| Antibody | Anti-RFP (Rabbit polyclonal) | Abcam | Cat. #: ab183628 | 1:5000 |
| Antibody | Anti-Flag (mouse monoclonal) | Sigma | F1804-200UG | 1:5000 |
| Antibody | Anti-His (mouse monoclonal) | TransGen Biotech | Cat. #: HT501 | 1:5000 |
| Antibody | Anti-MBP (mouse monoclonal) | ABclonal | Cat. #: AE016 | 1:5000 |
| Antibody | Anti-Histone H3 (Rabbit polyclonal) | Abcam | Cat. #: ab1791 | 1:5000 |
| Antibody | Anti-Actin (mouse monoclonal) | Affinity | Cat. #: AF7018 | 1:5000 |
| Antibody | Anti-Tubulin (Rabbit polyclonal) | PTM Biolabs | Cat. #: PTM-1011 | 1:2500 |
| Antibody | Anti-ubiquitination (mouse monoclonal) | Santa Cruz | Cat. #: sc-8017 | 1:2500 |
| Antibody | Anti-K167ac (Rabbit polyclonal) | HUABIO | Materials and methods | 1:10,000 |
| Recombinant DNA reagent | pTX041 (plasmid) | *Deng et al., 2018* | | |
| Recombinant DNA reagent | pQB-V3 (plasmid) | *Chen et al., 2020* | | |
| Recombinant DNA reagent | pK7FWG2 (plasmid) | *Karimi et al., 2002* | | |
| Recombinant DNA reagent | pGWB554 (plasmid) | *Nakagawa et al., 2007* | | |
| Recombinant DNA reagent | pEarleyGate201-YN (plasmid) | *Chen et al., 2020* | | |
| Recombinant DNA reagent | pEarleyGate202-YC (plasmid) | *Chen et al., 2020* | | |
| Sequence-based reagent | Primers | This paper | | *Supplementary file 1c* |
| Sequence-based reagent | Synthetic genes | This paper | | *Supplementary file 1d* |
| Commercial assay or kit | ClonExpress II One Step Cloning Kit | Vazyme | Cat. #: C112-01/02 | |
| Commercial assay or kit | Nuclear Extraction Kit | Solarbio | Cat. #: R0050 | |
| Commercial assay or kit | PrimeScriptTM RT reagent Kit | Takara | Cat. #: RR037A | |
| Commercial assay or kit | SYBR Premix Ex Taq | Mei5bio | Cat. #: MF013-plus-10 | |
| Commercial assay or kit | Omni-Easy Instant BCA Protein Assay Kit | Epizyme | Cat. #: ZJ102 | |
| Chemical compound, drug | Protease inhibitor mixture | Solarbio | Cat. #: P6730 | |
| Chemical compound, drug | Proteinase inhibitor cocktail | Sigma | Cat. #: P9599-5ML | |
| Chemical compound, drug | Anti-GFP agarose | KT Health | Cat. #: KTSM1301 | |
| Chemical compound, drug | Hygromycin B | Thermo Fisher Scientific | Cat. #: 10687010 | |
| Chemical compound, drug | G418 | Solarbio | Cat. #: IG0010 | |

*Continued on next page*

*Continued*

| Reagent type (species) or resource | Designation | Source or reference | Identifiers | Additional information |
|---|---|---|---|---|
| Chemical compound, drug | DAPI | Sigma | Cat. #: D9542 | |
| Software, algorithm | ImageJ software | ImageJ(http://imagej.nih.gov/ij/) | | Version 1.48 |
| Software, algorithm | GraphPad Prism software | GraphPad Prism (https://graphpad.com) | | Version 6.0.0 |

## Fungal strains, plants, and culture conditions

*F. oxysporum* f. sp. *lycopersici* strain *Fol* 4287 (*Ma et al., 2013*; *Ma et al., 2010*) was used in all experiments. The fungus was cultured on potato dextrose agar plates at 25°C to generate conidia. The harvested conidia at the optimum concentration of $5×10^6$ conidia/ml (*Vitale et al., 2019*) were inoculated in liquid minimal YEPD (0.03% yeast extract, 0.1% peptone, 0.2% dextrose) medium to induce conidial germination and secreted protein production (*Li et al., 2020a*). Cultures were grown at 25°C with shaking at 200 rpm to harvest mycelia for protein extraction. Tomato (*Solanum lycopersicum* cv. Alisa Craig [AC]) plants were grown in tissue culture bottles with 1/2 MS medium for root collection. Tomato and tobacco (*N. benthamiana*) seedlings were potted in a soil mix (vermiculite:humus = 1:2) and placed in a climatized greenhouse at 28°C with 65% relative humidity and a 16 hr photoperiod (*Cao et al., 2018*).

## Constructs for gene deletion, complementation, and overexpression in *Fol*

To generate a *FolSvp1* gene deletion mutant, two 0.7 kb fragments flanking the target gene were PCR-amplified with the primer pairs FolSvp1-up-F/R and FolSvp1-down-F/R. Thereafter, the two flanking sequences were linked with the hygromycin resistance cassette HPH by overlap PCR using the primer pair Hph-F/R. The 2.8 kb fragment was amplified using primer pairs FolSvp1-k-F/R and purified for protoplast transformation. For complementation, a genomic region encompassing the entire *FolSvp1* coding sequence (948 bp) and its native promoter (upstream 1.5 kb) sequence was amplified using the primers FolSvp1-C-F/R and inserted into the *Xho* I-digested pYF11 (G418 resistance) vector using the ClonExpress II One Step Cloning Kit (Vazyme). For site-directed mutagenesis of *FolSvp1*, the variants with changes at the K152, K167, K258, or K284 residue were generated with the primers listed in *Supplementary file 1c*. Constructs with the native gene promoter were transformed into Δ*FolSvp1*-KO7 protoplast.

To overexpress FolSvp1-GFP and its point mutants in *Fol*, we amplified *FolSvp1* using the primers FolSvp1-OE-F and FolSvp1-C-R and inserted into the pYF11 vector carrying the strong RP27 promoter. To overexpress FolArd1-Flag in *Fol*, the *FolArd1* gene was amplified using the primers FolArd1-Flag-F/R and inserted into the *Xho* I-digested pHZ126 vector (hygromycin resistance) carrying C-terminal Flag tag under the RP27 promoter. Then, the constructs were used for the protoplast transformation of WT *Fol* or FolSVP1-GFP overexpression strains. The transformants were confirmed by PCR, Western blotting, or microscopic observation of the fluorescent signal using an Olympus BX53 fluorescence microscope (Melville, NY, USA) to observe GFP and DAPI (D9542, Sigma), a widely used fluorescent label for the nucleus. All the primers used in this study were listed in *Supplementary file 1c*.

## Functional verification of FolSvp1 SP

The predicted N-terminal 19-amino acid SP sequences of FolSvp1 were fused in frame to the invertase gene in the pSUC2 vector by two complementary primer sequences (*Supplementary file 1c*). *Eco*RI and *Xho*I restriction enzymes were used to insert the SP sequences into the pSUC2 vector (*Yin et al., 2018*). The recombinant plasmids were then transformed into invertase secretion deficient yeast strain YTK12 by lithium acetate-mediated transformation. The function of the SP can be evaluated by using different selective media and color reaction to verify the secretion of invertase as described previously (*Yin et al., 2018*).

## Construction of binary vectors for agroinfiltration

For the subcellular localization of FolSvp1 in planta, the entire *FolSvp1* gene containing the sequence encoding the native SP was PCR-amplified using primer pairs FolSvp1-pQB-F/R. The forward primer ^ΔSP^FolSvp1-pQB-F was used to generate truncated *FolSvp1* lacking the sequence encoding the SP (^ΔSP^FolSvp1-GFP). The forward primer ^ΔSP+PR1SP^FolSvp1-pQB-F generated an artificial form of *FolSvp1* (^ΔSP+PR1SP^FolSvp1-GFP) containing the tomato SlPR1 SP (1–24 amino acids) instead of its endogenous SP (1–19 amino acids). FolSvp1-GFP point mutation variants and the nls1 and nls2 variants with changes in the two predicted nuclear localization motifs (NLS1 and NLS2) were generated by site-specific mutagenesis with the primers listed in *Supplementary file 1c*. All of the constructed plasmids were cloned into vector pQB-V3 (*Chen et al., 2020*). These constructs were transferred to the destination vector pK7FWG2 (*Karimi et al., 2002*) for C-terminal GFP fusion using Gateway LR Clonase Enzyme Mix (Invitrogen).

For the SlPR1 constructs, full-length *SlPR1* was PCR-amplified with the primer pairs SlPR1-pQB-F/R. An SlPR1 construct lacking the sequence encoding the SP (^ΔSP^SlPR1) was obtained by PCR using primers ^ΔSP^SlPR1-pQB-F and SlPR1-pQB-R. To fuse the NLS peptide to ^ΔSP^SlPR1, the forward primers (*Supplementary file 1c*) were extended with sequences encoding the NLS peptide (*Du et al., 2015*). For the SlChi3 constructions, the *SlChi3* was amplified with the primer pairs SlChi3-pQB-F/R. The PCR products were cloned into pQB-V3 and then recombined into the RFP-tagged pGWB554 vector (*Nakagawa et al., 2007*) or the GFP-tagged pK7FWG2 vector. To generate the BiFC system, the pQB-V3 constructs were recombined into the pEarleyGate201-YN and pEarleyGate202-YC vectors (*Chen et al., 2020*). For the PR1 homologs and truncated PR1 constructions, these genes were colony to pQB-V3 vector from their pGADT7 constructions, and then recombined into the pEarleyGate202-YC vectors with the primers in *Supplementary file 1c*. To test whether SlPR1 contribute to tomato resistance, we tried to induce SlPR1 mutation by CRISPR-cas9 system as described previously (*Deng et al., 2018*). Target sites for the *SlPR1* gene were designed based on the online tool CRISPR-P 2.0 (http://cbi.hzau.edu.cn/crispr/). Two single-guide RNAs were designed and integrated into pTX041 vector (*Deng et al., 2018*) to produce tomato transformants by primers pair (*Supplementary file 1c*). All the binary constructs were then transformed into *Agrobacterium* LBA4404.

## *Agrobacterium*-mediated transformation of *N. benthamiana* and tomato

The *Agrobacterium* transformants were used for the transient transformation of *N. benthamiana* as described previously (*Li et al., 2018b*). A strain containing the silencing suppressor P19 was co-infiltrated at an OD600 of 0.3. For BiFC assays, *Agrobacterium* strains harboring either the FolSvp1-nYFP or SlPR1-cYFP construct were mixed at a 1:1 ratio and co-transformed into *N. benthamiana* using a transient expression assay as described above. After inoculation, the leaves were incubated at 25°C in the greenhouse for 40–72 hr depending on the purpose. *N. benthamiana* transgenic plants expressing RFP:H2B were used to label plant nuclei (*Martin et al., 2009*). For plasmolysis, leaf tissues were cut and incubated for 6 hr with 800 mM mannitol and then imaged. To observe fluorescence, the infiltrated tobacco leaves were imaged under a fluorescence microscope as above.

To generate SlPR1-overexpressing transgenic tomato, the *Agrobacterium*-mediated transformation of tomato was performed following a previously described method (*Du et al., 2020*). Transformants were selected on medium containing 50 mg/l kanamycin. Putative transformants were confirmed by PCR and Western blotting. Similar method was used to generate transgenic tomato lines carrying SlPR1 variants.

## Protein extraction

The *Fol*-infected tissues or plants were frozen and ground in liquid nitrogen and then homogenized in protein extraction buffer as described previously (*Xian et al., 2020*). The culture supernatant protein was extracted as described previously (*Li et al., 2020a*). To determine that FolSvp1 could translocate to host roots, tomato roots were inoculated with a spore suspension ($5 \times 10^6$ spores/ml) of the FolSvp1-GFP-overexpressing transformants for 24 hr, and subsequently used for protoplast collection and protein extraction as described previously (*Ji et al., 2021*). Plant cytoplasm or nuclear proteins were extracted from agroinfiltrated *N. benthamiana* leaves using a Nuclear Extraction Kit (Solarbio) and a protease inhibitor mixture (Solarbio) 3 days after infiltration following the manufacturer's instructions.

Apoplastic fluid from agroinfiltrated leaves was isolated as described previously (*Dong et al., 2014*; *Shabab et al., 2008*).

## Western blotting

The obtained proteins were separated by 12% SDS-PAGE gel and immunoblotted using anti-GFP (1:5000 dilution, Abcam, ab183734), anti-RFP (1:5000 dilution, Abcam, ab183628), anti-Flag (1:5000 dilution, Sigma, F1804-200UG), anti-His (1:5000 dilution, TransGen Biotech, HT501), anti-MBP (1:5000 dilution, ABclonal, AE016), anti-Histone H3 (1:5000 dilution, Abcam, ab1791), anti-Actin (1:5000 dilution, Affinity, AF7018), anti-Tubulin (1:2500 dilution, PTM Biolabs, PTM-1011), and anti-ubiquitination (1:2500 dilution, Santa Cruz, sc-8017). The FolSvp1 K167 site-specific acetylation antibody, anti-K167ac, was generated by using a FolSvp1 acetylated peptide (KNDHDEDK(ac)ECVCFN) conjugated to KLH as an antigen. Antibodies were produced from rabbits by HUABIO (Hangzhou, China). The specificity of the antibody was tested by immunoblot analysis (1:10,000 dilution). The signals in the blots were visualized with ECL kits (Millipore) and photographed using an Amersham Imager 680 (GE Healthcare). SDS-PAGE was performed with silver staining or Coomassie brilliant blue staining to verify equal loading.

## IP and mass spectrometry

IP assays for LC-MS/MS analysis were performed as previously described (*Zhang et al., 2021*) with several modifications. FolSvp1-GFP overexpression *Fol* tissues (0.5 g) or SlPR1-GFP overexpression plant tissues (1 g) were collected for protein extraction as described above. GFP-Trap agarose beads (50 µl, KTSM1301, KT-HEALTH, China) were added to the clean protein extracts, which were then incubated for 2 hr at 4°C with end-to-end slow, constant rotation. GFP-Trap beads were washed three times with 1 ml of cold wash buffer (100 mM Tris-HCl pH 8, 150 mM NaCl, 10% glycerol, 2 mM DTT, 0.1% protease inhibitor cocktail [Sigma], 0.1% Triton X-100) and two additional times with wash buffer without Triton X-100 detergent. The washed beads were boiled with 50 µl SDT (4% SDS, 100 mM Tris-HCl, 1 mM DTT, pH 7.6), and the pulled down protein was subsequently subjected to SDS-PAGE, Western blotting, or LC-MS/MS.

## Co-IP assay

For the Co-IP assay, samples were collected from *Fol* co-expressing FolSvp1-GFP and FolArd1-Flag strains. Tissue powder samples (300 mg each) were suspended in 1 ml of cold extraction buffer for protein extraction as described above. GFP-tagged FolSvp1 fusions were immunoprecipitated with anti-GFP agarose beads, and the captured proteins were eluted from the beads as described above. The eluted proteins were separated by SDS-PAGE and subjected to immunoblot analysis.

## Y2H analysis

To generate the Y2H system, the coding sequence of FolSvp1 (without the SP, $^{\Delta SP}$FolSvp1) was cloned to the bait vector pGBKT7, which was then transformed into the Y2H Gold strain. A tomato Mate & Plate library was constructed and used for screening with the Matchmaker Gold Yeast Two-Hybrid System (Clontech) according to the manufacturer's instructions. For further verification of FolSvp1 candidate targets, the coding sequences of SlPR1 or SlMit were cloned into the prey vector pGADT7. Proper combinations of two of these plasmids were co-transformed into the Y2H Gold strain. The positive clones were collected and assayed for growth on DDO (SD/-Leu/-Trp) and QDO/A (SD/-Ade/-His/-Leu/-Trp with 125 ng/ml AbA) plates. Q or R variants at the K167 residue ($^{\Delta SP}$FolSvp1$^Q$ and $^{\Delta SP}$FolSvp1$^Q$) were introduced into pGBKT7, and Y2H assays were performed as described above. The coding sequences of 10 up-regulated tomato PR-1 proteins and truncated SlPR1 proteins were synthesized and introduced in pGADT7 by Tsingke Biotech, China. The full gene sequences were shown in *Supplementary file 1d*. All the constructs in Y2H assay removed its N-terminal predicted secretory SP according to SignalP sever.

## Heterologous expression and purification

*SlPR1* and *SlChi3* sequences lacking the N-terminal SP were cloned into the pExp-his-xMBP-TEV expression vector via the overhangs added by the primers during amplification (*Supplementary file 1c*). The His-MBP-SlPR1/SlChi3 fusion proteins were expressed in *Escherichia coli* BL21 cells

and purified as described previously (*Gamir et al., 2016*) with appropriate modifications. Cells were resuspended in lysis buffer containing 20 mM Tris-HCl, 500 mM NaCl, 20 mM imidazole, and 1 mM phenylmethylsulfonyl fluoride and lysed with a microfluidizer. N-terminal His-tagged SlPR1 and SlChi3 proteins were purified using Ni-nitrilotriacetic acid affinity resin (Qiagen) following the manufacturer's instructions. Bound proteins were eluted with a solution of 300 mM imidazole, 500 mM NaCl and 20 mM Tris-HCl, pH 8, and the eluted fraction was desalted on a Sephadex G-25 column (PD-10, GE Healthcare) equilibrated with 50 mM Tris-HCl, pH 8. Protein concentrations were determined with an Omni-Easy Instant BCA Protein Assay Kit (Epizyme) and subsequently visualized by Coomassie-stained SDS-PAGE before being concentrated using a 10 kDa MWCO Amicon Ultra centrifugal filter (Merck Millipore) to appropriate concentrations for further analysis.

*FolArd1* sequence was amplified using the primers FolArd1-His-F/R (*Supplementary file 1c*). This fragment was then cloned into a *Bam*H I-digested pET-28a expression vector via the overhangs added by the primers during amplification. The FolArd1-His protein was expressed in *E. coli* BL21 cells and purified as described above for the acetylation modification essay.

## In vitro acetylation assay

The in vitro acetylation essay was modified from a published method (*Song et al., 2016*). Briefly, the purified 5 µg FolArd1-His and 10 µg WT and K167 mutant FolSvp1-GFP proteins purified using anti-GFP agarose beads were incubated in the buffer (50 mM Tris-HCl pH 8.0, 10% glycerol, 0.1 mM EDTA, 1 mM dithiothreitol, 1 mM sodium butyrate) with 0.2 mM acetyl-CoA at 37°C for 1 hr. Then the reaction products were analyzed by Western blot.

## In vitro protein uptake assay

The protein uptake assay was performed using a published method (*Kale et al., 2010*) with small alterations. In brief, tomato roots were used for protoplast collection as described above. Fifty µg purified SlPR1-MBP or SlChi3-MBP protein was incubated with protoplast in the presence of GFP or FolSvp1-GFP proteins pulled down from *Fol* for 4 hr. After uptake, the mixture was washed five times with 1 ml of water. Then, nuclear proteins were extracted with the Nuclear Extraction Kit (Solarbio) and analyzed by Western blotting.

## Disease assays

Pathogenicity tests were performed using the modified root dip method (*Di et al., 2016*; *Zhang et al., 2021*). In brief, 14-day-old tomato seedlings were uprooted from the soil, inoculated with a spore suspension (5×10^6 spores/ml) of *Fol* or mock-treated (no spores) for 10 min, and subsequently potted and kept at 25°C. Two weeks later, disease symptoms of 8–20 plants/treatment were scored using a disease index ranging from 0 to 5 (0, no symptoms; 1, 0–20% defoliated leaves; 2, 20–40% defoliated leaves; 3, 40–60% defoliated leaves; 4, 60–80% defoliated leaves; and 5, 80–100% defoliated leaves).

To demonstrate that C-terminal CAPE1 peptide can enhance tomato resistance, two groups of tomato plants were presprayed with water or synthetic CAPE1 for 8 hr as described previously (*Chen et al., 2014*). Then, the tomato seedlings were inoculated with the *Fol* conidia for disease assays.

## RNA extraction for qRT-PCR analysis

For qRT-PCR essay, *Fol*-infected tomato roots were harvested at 0–5 days post inoculation. Infection was performed as follows: the harvested microconidia were suspended in 100 ml of water at a concentration of 5×10^6 spores/ml; 14-day-old tomato seedlings (40–50 seedlings/spot) were irrigated with the suspended microconidia and subsequently kept at 25°C. Total RNA was isolated using TRIzol reagent (Invitrogen) according to the manufacturer's instructions. Total RNA (2 µg) was used for reverse transcription with the PrimeScriptTM RT reagent Kit (TaKaRa). Quantitative expression assays were performed by using the 2× M5 HiPer SYBR Premix EsTag kit (Mei5bio) and a Light-Cycle96 (Roche) real-time PCR detection system. The relative quantification of gene expression was performed using the $2^{-\Delta\Delta ct}$ method (*Li et al., 2018a*). Data were normalized against the *SlActin* gene (NC_015448.3). The primer pairs used for real-time PCR are listed in *Supplementary file 1c*.

## Bioinformatic analyses

SP prediction was performed using the SignalP 5.0 server (http://www.cbs.dtu.dk/services/SignalP/). To generate the phylogenetic tree and sequence alignments, Blastp was used to search for homologs

of FolSvp1 in the NCBI database (https://www.ncbi.nlm.nih.gov/). Only hits with an E-value $<1e^{-60}$ for which the alignment returned by BLAST spanned more than 60% of both the query and the subject sequence were included. The phylogenetic tree was generated using the neighbor-joining method in MEGA-X. Multiple alignments were conducted using DNAMAN 6 software (Lynnon Biosoft, San Ramon, CA, USA). The online tool MEME (*Bailey et al., 2009*) was used to identify the potential conserved motifs of FolSvp1. The prediction of importin α-dependent NLSs was performed using the cNLS Mapper (*Kosugi et al., 2009b*). Statistical analysis was performed by analysis of variance (ANOVA) or by unpaired Student's t-test methods.

## Acknowledgements

This research was financially supported by the National Natural Science Foundation of China (31901830, 31972213), the Natural Science Foundation of Shandong Province (ZR2019BC032 and ZR2020KC003), the Ministry of Agriculture of China (2016ZX08009003-001), the Key Research and Development Program of Shandong Province (2019YQ017), Shandong Province 'Double-Hundred Talent Plan' (WST2018008), and Taishan Scholar Construction Foundation of Shandong Province (tshw20130963). Ben F Luisi was supported by the Wellcome Trust Investigator awards (200873/Z/16/Z, 222451/Z/21/Z).

## Additional information

### Funding

| Funder | Grant reference number | Author |
| --- | --- | --- |
| National Natural Science Foundation of China | 31901830 | Jingtao Li |
| National Natural Science Foundation of China | 31972213 | Wenxing Liang |
| Natural Science Foundation of Shandong Province | ZR2019BC032 | Jingtao Li |
| Natural Science Foundation of Shandong Province | ZR2020KC003 | Wenxing Liang |
| Ministry of Agriculture of China | 2016ZX08009003-001 | Wenxing Liang |
| Key Research and Development Program of Shandong Province | 2019YQ017 | Wenxing Liang |
| Shandong Province "Double-Hundred Talent Plan" | WST2018008 | Wenxing Liang |
| Taishan Scholar Project of Shandong Province | tshw20130963 | Wenxing Liang |
| Wellcome Trust | Investigator 200873/Z/16/Z | Ben F Luisi |
| Wellcome Trust | Investigator 222451/Z/21/Z | Ben F Luisi |

The funders had no role in study design, data collection and interpretation, or the decision to submit the work for publication. For the purpose of Open Access, the authors have applied a CC BY public copyright license to any Author Accepted Manuscript version arising from this submission

### Author contributions

Jingtao Li, Conceptualization, Resources, Data curation, Formal analysis, Supervision, Funding acquisition, Validation, Investigation, Visualization, Methodology, Writing - original draft, Project administration, Writing - review and editing; Xiaoying Ma, Chenyang Wang, Data curation,

Formal analysis, Methodology; Sihui Liu, Formal analysis, Investigation, Methodology; Gang Yu, Resources, Formal analysis, Methodology; Mingming Gao, Investigation, Methodology; Hengwei Qian, Mengjie Liu, Resources, Methodology; Ben F Luisi, Resources, Methodology, Writing - review and editing; Dean W Gabriel, Methodology, Writing - review and editing; Wenxing Liang, Conceptualization, Resources, Data curation, Formal analysis, Supervision, Funding acquisition, Validation, Investigation, Methodology, Writing - original draft, Project administration, Writing - review and editing

### Author ORCIDs
Jingtao Li  http://orcid.org/0000-0003-2702-1736
Gang Yu  http://orcid.org/0000-0002-6720-7490
Ben F Luisi  http://orcid.org/0000-0003-1144-9877
Wenxing Liang  http://orcid.org/0000-0002-3791-4901

### Decision letter and Author response
Decision letter https://doi.org/10.7554/eLife.82628.sa1
Author response https://doi.org/10.7554/eLife.82628.sa2

## Additional files

### Supplementary files
• Supplementary file 1. Mass spectrum data, primers and synthesized genes used in this study. (a) Target proteins obtained from the *Fol*-Secreted Virulence-related Protein1 (FolSvp1)-GFP co-purified liquid chromatography-tandem mass spectrometry (LC-MS/MS) data. (b) Target proteins obtained from the GFP co-purified LC-MS/MS data. (c) Primers used in this study. (d) Synthesized genes used in yeast two-hybrid (Y2H) study.

• MDAR checklist

### Data availability
All data generated or analysed during this study are included in the manuscript and supporting file.

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
