## [Editor Report]

The authors provided strong evidence that the Fusarium oxysporum effector protein FolSpv1 enhances virulence by targeting tomato SlPR1 and preventing the generation of the SlPR1-derived phytocytokine CAPE1, which otherwise positively regulates disease resistance in tomato plants. Strikingly, they show that FolSpv1 translocates SlPR1 from the apoplast back into the nucleus of tomato cells, suggesting a previously unknown mechanism employed by pathogenic microbes.

---

## [Decision Letter]

**Decision letter after peer review:**

Thank you for submitting your article "Acetylation of a fungal effector that translocates host PR1 facilitates virulence" for consideration by *eLife*. Your article has been reviewed by 2 peer reviewers, including Jian-Min Zhou as Reviewing Editor and Reviewer #1, and the evaluation has been overseen by Detlef Weigel as the Senior Editor. The following individual involved in review of your submission has agreed to reveal their identity: Jianfeng Li (Reviewer #2).

Essential revisions:

The claim that FolSpv1-mediated translocation of SlPR1 into the nucleus impedes CAPE1 release needs to be further strengthened with additional data. Otherwise the claim needs to be toned down.

*Reviewer #1 (Recommendations for the authors):*

The description of methods is not complete. How was in vitro protein uptake done? How was nuclear protein fractionation done?

*Reviewer #2 (Recommendations for the authors):*

1. Ideally, the authors may want to test whether the bacterially expressed SlPR1 proteins can impair the Fol growth.

2. It is better to testify the localization of FolSvp1-SlPR1 interaction by BiFC using FolSvp1-overexpression Fol strain and SlPR1-overexpression transgenic tomato plants, because this is more close to physiological conditions.

3. The FolArd1 overexpression Fol strain is expected to be more virulent if the authors' model is correct. It is better to verify this point.

4. "Western analyses" should be "Western blot analyses".

5. Figure 1F, why the R mutant of FolSvp1 shows an equal abundance as WT and the Q mutant?

6. Figure 4F, why the FolSvp1 has no basal acetylation after being pulled down.

7. Figure 5B, NLS2 in the diagram was mistakenly labeled as NLS1.

---

## [Author Response]

Reviewer #1 (Recommendations for the authors):The description of methods is not complete. How was in vitro protein uptake done? How was nuclear protein fractionation done?

We have added this information to the revised manuscript (lines 652-659).

Reviewer #2 (Recommendations for the authors):1. Ideally, the authors may want to test whether the bacterially expressed SlPR1 proteins can impair the Fol growth.

We have discussed this possibility and stated that additional experiments are required in future studies in the revised manuscript (lines 436-444).

2. It is better to testify the localization of FolSvp1-SlPR1 interaction by BiFC using FolSvp1-overexpression Fol strain and SlPR1-overexpression transgenic tomato plants, because this is more close to physiological conditions.

Thank you for this excellent suggestion. We have stated that the BiFC assays performed with *N. benthamiana* leaves might not completely mimic the physiological conditions and that new analyses using the *FolSvp1*-overexpression *Fol* strain and the *SlPR1*-overexpression tomato would greatly strengthen our conclusions in the revised manuscript (lines 439-444).

3. The FolArd1 overexpression Fol strain is expected to be more virulent if the authors' model is correct. It is better to verify this point.

As shown in Figures 1-3, FolSvp1 is largely acetylated in the presence of tomato roots, and we guess that *FolArd1* overexpression will have little effect on *Fol* virulence. We would like to verify this point in future studies.

4. "Western analyses" should be "Western blot analyses".

Changes have been made (lines 135 and 185).

5. Figure 1F, why the R mutant of FolSvp1 shows an equal abundance as WT and the Q mutant?

We loaded the same amount of the WT and mutant FolSvp1 proteins to compare their acetylation level.

6. Figure 4F, why the FolSvp1 has no basal acetylation after being pulled down.

Only in the presence of tomato roots, FolSvp1 is acetylated and then stabilized. To investigate the effect of FolArd1, we pulled down FolSvp1 lack of acetylation from mycelia in the absence of tomato roots. We have indicated this information in the figure legends (lines 1041-1042).

7. Figure 5B, NLS2 in the diagram was mistakenly labeled as NLS1.

Change has been made in the revised manuscript.